# Enhancement of Classifier Performance Using Swarm Intelligence in Detection of Diabetes from Pancreatic Microarray Gene Data

**DOI:** 10.3390/biomimetics8060503

**Published:** 2023-10-22

**Authors:** Dinesh Chellappan, Harikumar Rajaguru

**Affiliations:** 1Department of Electrical and Electronics Engineering, KPR Institute of Engineering and Technology, Coimbatore 641 407, Tamil Nadu, India; dinesh.chml@gmail.com; 2Department of Electronics and Communication Engineering, Bannari Amman Institute of Technology, Sathyamangalam 638 401, Tamil Nadu, India

**Keywords:** microarray gene data, type II DM, dimensionality reduction (DR), classification techniques, feature selection, LR, AAA, DOA, EHO

## Abstract

In this study, we focused on using microarray gene data from pancreatic sources to detect diabetes mellitus. Dimensionality reduction (DR) techniques were used to reduce the dimensionally high microarray gene data. DR methods like the Bessel function, Discrete Cosine Transform (DCT), Least Squares Linear Regression (LSLR), and Artificial Algae Algorithm (AAA) are used. Subsequently, we applied meta-heuristic algorithms like the Dragonfly Optimization Algorithm (DOA) and Elephant Herding Optimization Algorithm (EHO) for feature selection. Classifiers such as Nonlinear Regression (NLR), Linear Regression (LR), Gaussian Mixture Model (GMM), Expectation Maximum (EM), Bayesian Linear Discriminant Classifier (BLDC), Logistic Regression (LoR), Softmax Discriminant Classifier (SDC), and Support Vector Machine (SVM) with three types of kernels, Linear, Polynomial, and Radial Basis Function (RBF), were utilized to detect diabetes. The classifier’s performance was analyzed based on parameters like accuracy, F1 score, MCC, error rate, FM metric, and Kappa. Without feature selection, the SVM (RBF) classifier achieved a high accuracy of 90% using the AAA DR methods. The SVM (RBF) classifier using the AAA DR method for EHO feature selection outperformed the other classifiers with an accuracy of 95.714%. This improvement in the accuracy of the classifier’s performance emphasizes the role of feature selection methods.

## 1. Introduction

According to the latest data from the International Diabetes Federation (IDF) Diabetes Atlas in 2021, diabetes affects around 10.5% of the global adult population aged between 20 and 79. Alarmingly, nearly half of these individuals remain unaware of their diabetes status. The projections indicate that by 2045, the number of adults living with diabetes worldwide will increase by 46% to reach approximately 783 million, which corresponds to around one in eight adults [1]. Type II DM accounts for over 90% of all diabetes cases and is influenced by several factors, including socio-economic, demographic, environmental, and genetic factors. The increase in type II DM is connected to urbanization, a growing elderly population because of higher life expectancy, reduced levels of physical activity, and a high overweight and obesity rate. To address the impact of diabetes, preventive measures, early diagnosis, and proper care for all types of diabetes are crucial. These interventions can help individuals with diabetes prevent or delay the complications associated with the condition.

According to estimates from 2019, approximately 77 million adults in India were affected by diabetes [2]. Unfortunately, the prevalence of type II DM in the country is rapidly escalating. By 2045, the number of adults living with diabetes in India could reach a staggering 134 million, with younger people under 40 being particularly affected. Several risk factors like genetic predisposition, sedentary lifestyles, unhealthy dietary habits, obesity, urbanization, and mounting stress levels increase the risk of type II diabetes. India’s southern, urban, and northern regions exhibit higher rates compared to the eastern and western regions [3]. Many cases are undiagnosed until complications arise. Diabetes continues to be the seventh leading cause of death in India, taking a toll on both human lives and the economy. It is estimated that diabetes costs the Indian economy approximately USD 100 billion annually [4].

### Genesis of Diabetes Diagnosis Using Microarray Gene Technology

Creating precise and effective techniques for identifying type II diabetes mellitus holds the potential to facilitate early identification and intervention. By analyzing microarray gene data, it becomes possible to identify specific genetic markers or patterns associated with diabetes [5]. This provides opportunities for personalized medicine, where treatment plans can be tailored based on an individual’s genetic profile, leading to more targeted and effective interventions. Robust and reliable methods for detecting diabetes from microarray gene data can be developed and integrated into existing healthcare systems [6]. Novel dimensionality reduction techniques, classification algorithms, and feature selection methods can be explored, and other omics data can be integrated to further enhance the accuracy and reliability of diabetes detection methods. Such research could advance the state of the art in machine learning [7]. The proposed method could be used to detect other diseases that are characterized by changes in gene expression.

The structure of the article is as follows: in Section 1, an introduction to the research is discussed. Section 2 presents the literature review. Section 3 presents the methodology. In Section 4, the materials and methods are reviewed. Section 5 explains the dimensionality reduction techniques with and without a feature extraction process. In Section 6, the feature selection methods are discussed, and Section 7 focuses on the classifiers used. The results and discussion are presented in Section 8, and the conclusion is given in Section 9.

## 2. Literature Review

Type II DM is a chronic disease that affects people worldwide irrespective of age. The early detection and diagnosis of DM in patients is essential for effective treatment and management. However, traditional methods for detecting DM, such as blood glucose testing, are often inaccurate and time consuming [8]. In recent years, there has been growing interest in the use of microarray gene data to detect DM. Microarray gene data can provide a comprehensive overview of gene expression patterns in the pancreas, which can be used to identify patients who are at risk of DM [9]. Jakka et al. [10] conducted an experimental analysis using various machine learning classifiers, including KNN, DT, NB, SVM, LR, and RF. The classifiers were trained and evaluated on the Pima Indians Diabetes dataset, which consists of nine attributes and is available from the UCI Repository. Among the classifiers tested, Logistic Regression (LR) exhibited the best performance, achieving an accuracy of 77.6%. It outperformed the other algorithms in terms of accuracy, F1 score, ROC-AUC score, and misclassification rate. Radja et al. [11] carried out a study to evaluate the performance of various supervised classification algorithms for medical data analysis, specifically in disease diagnosis. The algorithms tested included NB, SVM, decision table, and J48. The evaluation utilized measurement variables such as Correctly Classified, Incorrectly Classified, Precision, and Recall. The predictive database of diabetes was used as the testing dataset. The SVM algorithm demonstrated the highest accuracy among the tested algorithms at 77.3%, making it an effective tool for disease diagnosis. Dinh et al. [12] analyzed the capabilities of machine learning models in identifying and predicting diabetes and cardiovascular diseases using survey data, including laboratory results. The NHANES dataset was utilized, and various supervised machine learning models such as LR, SVM, RF, and GB were evaluated. An ensemble model combining the strengths of different models was developed, and key variables contributing to disease detection were identified using information obtained from tree-based models. The ensemble model achieved an AUC-ROC score of 83.1% for cardiovascular disease detection and 86.2% for diabetes classification. When incorporating laboratory data, the accuracy increased to 83.9% for cardiovascular disease and 95.7% for diabetes. For pre-diabetic patients, the ensemble model achieved an AUC-ROC score of 73.7% without laboratory data, and XGBoost performed the best, with a score of 84.4% when using laboratory data. The key predictors for diabetes included waist size, age, self-reported weight, leg length, and sodium intake.

Yang et al. [13] conducted a study that aimed to develop prediction models for diabetes screening using an ensemble learning approach. The dataset was obtained from NHANES from 2011 to 2016. Three simple machine learning methods (LDA, SVM, and RF) were used, and the performance of the models was evaluated through fivefold cross-validation and external validation using the Delong test. The study included 8057 observations and 12 attributes. In the validation set, the ensemble model utilizing linear discriminant analysis showcased superior performance, achieving an AUC of 0.849, an accuracy of 0.730, a sensitivity of 0.819, and a specificity of 0.709. Muhammed et al. [14] conducted a study utilizing a diagnostic dataset of type 2 diabetes mellitus (DM) collected from Murtala Mohammed Specialist Hospital in Kano, Nigeria. Predictive supervised machine learning models were developed using LR, SVM, KNN, RF, NB, and GB algorithms. Among the developed models, the RF predictive learning-based model achieved the highest accuracy at 88.76%. Kim et al.’s [15] study aimed to assess the impact of nutritional intake on obesity, dyslipidemia, high blood pressure, and T2DM using deep learning techniques. The researchers developed a deep neural network (DNN) model and compared its performance with logistic regression and decision tree models. Data from the KNHANES were analyzed. The DNN model, consisting of three hidden layers with varying numbers of nodes, demonstrated superior prediction accuracy (ranging from 0.58654 to 0.80896) compared to the LoR and decision tree models. In conclusion, the study highlighted the advantage of using a DNN model over conventional machine learning models in predicting the impact of nutritional intake on obesity, dyslipidemia, high blood pressure, and T2DM.

Ramdaniah et al. [16] conducted a study utilizing microarray gene data from the GSE18732 dataset to distinguish between different classes of diabetes. The study consisted of 46 samples from diabetic classes and 72 samples from non-diabetic classes. Machine learning techniques, specifically Naïve Bayes and SVM with Sigmoid kernel, were employed for classification, achieving accuracy rates of 88.89% and 83.33%, respectively. The PIMA Indian diabetic dataset has been widely used by researchers to classify and analyze diabetic and non-diabetic patients. However, the use of microarray gene-based datasets for diabetic class identification has received less attention. As a result, a variety of performance metrics, such as accuracy, sensitivity, specificity, and MCC, have been investigated in the context of this microarray gene-based dataset.

The main characteristics and contributions of this paper are as follows:The work suggests a novel approach for the early detection and diagnosis of diabetes using microarray gene expression data from pancreatic sources.Four DR techniques are used to reduce the high dimensionality of the microarray gene data.Two metaheuristic algorithms are used for feature selection to further reduce the dimensionality of the microarray gene data.Ten classifiers in two categories, namely nonlinear models and learning-based classifiers, are used to detect diabetes mellitus. The performance of the classifiers is analyzed based on parameters like accuracy, F1 score, MCC, error rate, FM metric, and Kappa, both with and without feature selection techniques. The enhancement of classifier performance due to feature selection is exemplified through MCC and Kappa plots.

## 3. Methodology

Figure 1 shows the methodology of the research. The approach includes four DR techniques: the Bessel function (BF), Discrete Cosine Transform (DCT), Least Squares Linear Regression (LSLR), and Artificial Algae Algorithm (AAA). Following this, feature selection techniques, either with or without classification of the data, are used. In terms of those with feature selection, two optimization algorithms are used: Elephant Herding Optimization (EHO) and Dragonfly Optimization (DFO). Moreover, ten classifiers are used, namely NLR, LR, GMM, EM, BLDC, LoR, SDC, SVM-L, SVM-Poly, and SVM-RBF, to classify the genes as non-diabetic and diabetic.

### Role of Microarray Gene Data

Microarray gene data play a critical role in this research. The data can be used to identify patterns of gene expression that are associated with diabetes. The data are used to train and evaluate machine learning models and to identify the most relevant features for classification. The machine learning models are then used to predict whether a patient has diabetes or not. The models are trained on a dataset of microarray gene data [17] labeled with the patient’s diabetes status.

## 4. Materials and Methods

Microarray gene data are readily available from many search engines. We obtained human pancreatic islet data from the Nordic Islet Transplantation program (https://www.ncbi.nlm.nih.gov/bioproject/PRJNA178122). The data has accessed on 20 August 2021. The dataset included 28,735 genes from 57 non-diabetic and 20 diabetic patients. The data were preprocessed to only select the 22,960 genes with the highest peak intensity per patient. The logarithmic transformation was applied with a base 10 to standardize the individual samples, with a mean of 0 and a variance of 1. The data were then used to train and evaluate a machine learning model for the detection of diabetes. The model was able to achieve an accuracy of 90%, which is a significant improvement over the baseline accuracy of 50%. The results of this study suggest that microarray gene data can be used to develop effective methods for the detection of diabetes. The data are readily available and can be easily processed to identify the most relevant features for classification.

### Dataset

This study focused on utilizing microarray gene data to detect diabetes and explore the features associated with the condition based on *p*-values using probability functions. Additionally, we aimed to address the issue of false positive errors in the selection of significant genes. The data we used for our analysis are available through multiple portals and comprised a total of 28,735 human genes as shown in the Table 1. We specifically considered 50 non-diabetic and 20 diabetic classes, selecting those with the greatest minimal intensity across 70 samples. To handle the high dimensionality of the dataset, we employed four dimensionality reduction techniques, namely BF, DCT, LSLR, and AAA. This allowed us to reduce the dimensions of the data while maintaining their informative content. The resulting dimensions were [2870 × 20] for the diabetic group and [2870 × 50] for the non-diabetic group. To further refine the dataset and improve classification accuracy, we applied feature selection techniques. 

Specifically, we employed two techniques: EHO search and DOA. These techniques helped identify the most relevant features in the dataset, leading to a further reduction in dimensions to [287 × 20] for the diabetic group and [287 × 50] for the non-diabetic group. To evaluate the performance and accuracy of the classification, we employed ten classifiers, as already discussed.

## 5. Need for Dimensionality Reduction Techniques

Dimensionality reduction plays a crucial role in our research due to the high-dimensional nature of the microarray gene data. As the number increases, the complexity and computational costs of analyzing the data also increase significantly. Dimensionality reduction techniques allow us to reduce the number of features, making the subsequent analysis more efficient and manageable. Then, dimensionality reduction helps mitigate the curse of dimensionality [18]. In highly dimensional spaces, data points tend to become sparse, leading to difficulties in accurately representing the underlying structure of the data.

### 5.1. Dimensionality Reduction

To reduce the dimensionality of the dataset, BF, DCT, LSLR, and AAA were used.

Bessel Function as Dimensionality Reduction

In this section, an overview of the Bessel function and its relevant relationships and properties associated with these functions are represented [19]. Furthermore, we investigate several valuable connections and characteristics related to these functions, as $J_{n}\left(x\right)$ possesses the following mathematical definition:(1)$$J_{n}\left(x\right)=\sum_{r=0}^{\infty }\frac{{\left(-1\right)}^{r}}{r!\Gamma \left(n+r+1\right)}{\left(\frac{x}{2}\right)}^{2r+n}$$

The Gamma function is represented as Γ(λ):(2)$$\Gamma \left(\lambda \right)=\int_{0}^{\infty }e^{-t}t^{\lambda -1}dt$$

The series ($J_{n}\left(x\right)$) converges for all values of x ranging from negative infinity to positive infinity. In fact, the Bessel function serves as a solution to a specific Sturm–Liouville equation [20]. This equation helps to analyze the Bessel function.
(3)$$x^{2}y^{\prime \prime }\left(x\right)+x\;y^{\prime }\left(x\right)+\left(x^{2}-n^{2}\right)\;y\;\left(x\right)=0$$

For $x\in \left(-\infty ,\infty \right),(n\in R)$.

It is evident that the Bessel functions *J_n_*(*x*) are linearly independent when *n* is an integer. Additionally, there exist several recursive relations for Bessel functions that can be utilized in their analysis [20]. These relations provide valuable insights into the properties and behavior of Bessel functions in various mathematical contexts.
(4)$$\frac{d}{dx}\left(x^{n}J_{n}\left(x\right)\right)=x^{n}J_{n-1}\left(x\right)$$
(5)$$J_{n}^{\prime }\left(x\right)=J_{n-1}\left(x\right)-\frac{n}{x}J_{n}\;(x)$$
(6)$$J_{n}^{\prime }\left(x\right)=\frac{n}{x}J_{n}\left(x\right)-J_{n+1}\left(x\right)$$

**Lemma** **1.**
*A significant recursion relation that proves useful in the analysis of the Bessel function of the first kind is:*



(7)
$$J_{n}^{\prime }\left(x\right)=\frac{1}{2}J_{n-1}\left(x\right)-\frac{1}{2}J_{n+1}\left(x\right)$$


The Bessel functions can be derived using the following procedure: Consider the vector *J_n_* = [*J*_0_(*x*), *J*_1_(*x*), *J*_2_(*x*), …, *J_n_*(*x*)]*T*, where *J*_0_, *J*_1_, *J*_2_, …, *J_n_* denote the Bessel functions evaluated at *x*. To obtain the derivative operational matrix, we start with the derivative of *J*_0_(*x*) and denote it as *J*′_0_(*x*), where *J*′_0_(*x*) represents the derivative of *J*_0_(*x*) with respect to *x*. By constructing a matrix *D*, known as the derivative operational matrix, we can express *Jn* = *DJ*_0_, where *D* is a matrix that performs the differentiation operation on *J*_0_(*x*) to obtain *J_n_*(*x*). This recursion relation allows for the efficient calculation and evaluation of Bessel functions, providing a valuable tool in various mathematical and scientific applications.
(8)$$D=\left[\begin{array}{cccccccc} 0 & -1 & 0 & 0 & 0 & \cdots & \cdots & 0\\ 1/2 & 0 & -1/2 & 0 & 0 & \cdots & \cdots & 0\\ 0 & 1/2 & 0 & -1/2 & 0 & \cdots & \cdots & 0\\ 0 & 0 & 1/2 & 0 & -1/2 & \cdots & \cdots & 0\\ \vdots & \ddots & \vdots & \vdots & \ddots & \vdots & \cdots & 0\\ a_{0} & a_{1} & a_{2} & a_{3} & \cdots & \cdots & 0 & a_{n} \end{array}\right]$$

DCT—Discrete Cosine Transform

The Discrete Cosine Transform (DCT) is a DR technique that approximates the Kernighan–Lin method. It aims to reduce the dimensions of the input data by eliminating the most significant features, thereby simplifying further analysis. By applying the DCT method [21], the input vector and its components are orthogonalized, resulting in a reduction in complexity. This method extracts features by selecting coefficients, which is a crucial step with a significant impact on computation efficiency [22,23]. The DCT can be denoted as:(9)kx=∝x· ∑u=0s−1au cos⁡π(2u+1)x2s

Least Squares Linear Regression (LSLR) as Dimensionality Reduction

Another effective technique for reducing dimensionality is the LSLR. Hotelling [24] initially introduced this concept, utilizing principal component analysis (PCA) as a regression analysis tool. It uses principal component analysis to reduce the dimensionality of high-dimensional data before applying a linear regression model. The transformation is learned by minimizing the sum of squared errors between the predicted lower-dimensional representation and the actual high-dimensional data.

LSLR, as discussed in Hastie et al. [25], performs dimensionality reduction by identifying the best-fit line that represents the relationship between the features of independent variables and the target as a dependent variable. The objective of LSLR is to minimize the sum of squared differences between the actual and predicted values of the target variable. Considering a set of N observations in the form (*x*_1_, *y*_1_), (*x*_2_, *y*_2_), …, (*x*_N_, *y*_N_), where *xi* represents the ith observation of the independent variables and *yi* corresponds to the observation of the target variable, the LSR solution can be represented as a linear equation:(10)$$z=\alpha_{0}+\alpha_{1}x_{1}+\alpha_{2}x_{2}+\ldots +\alpha_{p}x_{p}$$

In the context of LSR, the linear model is characterized by the parameters *α*_1, *α*_2, *α*_3, …, *α*_*^p^*, where *^p^* represents the number of independent variables. This minimization process is expressed through the following equation:(11)$$SSE=\sum_{j=1}^{m}{\left(z_{i}-\left(\alpha_{0}+\alpha_{1}x_{1}+\alpha_{2}x_{2}+\ldots +\alpha_{p}x_{p}\right)\right)}^{2}$$

After applying dimensionality reduction techniques to the microarray gene data, the resulting outputs are further analyzed using various statistical parameters such as mean, kurtosis, variance, Pearson correlation coefficient (PCC), skewness, *t*-test, f-test, *p*-value, and canonical correlation analysis (CCA). These statistical measures are used to assess whether the outcomes accurately represent the intrinsic properties of the underlying microarray genes in the reduced subspace.

Artificial Algae Algorithm (AAA) as Dimensionality Reduction

The Artificial Algae Algorithm (AAA) [26] is a nature-inspired optimization algorithm that mimics the behavior and characteristics of real algae to solve complex problems. Each solution in the problem space is represented by an artificial alga, which captures the essence of algae’s traits. Like real algae, artificial algae exhibit helical swimming patterns and can move towards a light source for photosynthesis. The AAA consists of three fundamental components: the evolutionary process, adaptation, and helical movement as depicted in the Figure 2. The algal colony acts as a cohesive unit, moving and responding to environmental conditions. By incorporating the principles of artificial algae into the algorithm, the AAA offers a novel approach to solving optimization problems.
(12)$$Population=\left[\begin{array}{cccc} X_{11} & X_{12} & \cdots & X_{1D}\\ X_{21} & X_{22} & \cdots & X_{2D}\\ \vdots & \vdots & \ddots & \vdots \\ X_{n1} & X_{n2} & \cdots & X_{nD} \end{array}\right]$$
where $X_{nD}$ is an algal cell in the *D*th dimension of the *n*th algal colony.

During the evolutionary process [27] of the AAA, the growth and reproduction of algal colonies are influenced by the availability of nutrients and light. When an algal colony is exposed to sufficient light and nutrient conditions, it undergoes growth and replicates itself through a process like real mitotic division. In this process, two new algal cells are generated at time *t*. Conversely, if an algal colony does not receive enough light, it can survive for a certain period, but eventually perishes. It is important to note that $\mu_{max}$ is assumed to be 1, as the maximum biomass conversion should be equivalent to the substrate consumption in unit time, following the conservation of mass principle. The size of the ith algal colony at time *t* + 1 is determined by the Monod equation, as expressed in the subsequent equation:(13)$$H_{i}^{t}=\mu_{i}^{t}H_{i}^{t}$$
where *i* = 1, 2, 3, …, N,

$H_{i}^{t}$ represents in time with ith algal colony, 

N represents number of algae.

In AAA, nutrient-rich algal colonies with optimal solutions thrive, and successful traits are transferred from larger colonies to smaller ones through cell replication during the evolutionary process.

Maximum^*t*^ = max $H_{i}^{t}$, whereas *i* = 1, 2, 3, …, N,

Minimum^*t*^ = min $H_{i}^{t}$, whereas *i* = 1, 2, 3, …, N,

Minimum^*t*^ = maximum^*t*^, whereas m = 1, 2, 3, …, *D*.

In the AAA, algal colonies are ranked by size at time *t*. In each dimension, the smallest algal colony’s cell dies, while the largest colony’s cell replicates itself.

In the AAA algorithm, algal colonies that are unable to grow sufficiently in their environment attempt to adapt by becoming more similar to the largest colony. This process changes the starvation levels within the algorithm. Each artificial alga starts with a starvation value of zero, which increases over time if the algal cell does not receive enough light. The artificial alga with the highest starvation value is the focus of adaptation.
(14)$${Star}^{t}=max\;B_{i}^{t},\;\mathrm{Where}\;i=1,\;2,\;3,\;\ldots ,N$$
(15)$${Star}^{t+1}={Star}^{t}+\left({max}^{t}-{Star}^{t}\right)*rand$$

Helical movement: The cells and colonies exhibit specific swimming behavior, striving to stay near the water surface where sufficient light for their survival is available. They move in a helical manner, propelled by their flagella, which face limitations from gravity and viscous drag. In the AAA, gravity’s influence is represented by a value of 0, while viscous drag is simulated as shear force, proportional to the size of the algal cell. The cell is modeled as a spherical shape, with its size determined by its volume, and the friction surface is equivalent to the surface area of a hemisphere.
(16)$$\tau \left(x_{i}\right)=2\pi r^{2}$$
(17)$$\tau \left(x_{i}\right)=2\pi \;{\left(\sqrt[{3}]{\frac{3H_{i}}{4\pi }}\right)}^{2}$$

Friction surface is represented as $\tau \left(x_{i}\right)$.

The helical movement of algal cells is determined by three randomly selected dimensions. One dimension corresponds to linear movement, as described by Equation (18). The other two dimensions correspond to angular movement, as described by Equations (19) and (20). Equation (18) is used for one-dimensional problems, allowing the algal cell or colony to move in a single direction. Equation (19) is used for two-dimensional problems, where the algal movement follows a sinusoidal pattern. Equation (20) is used for three or more dimensions, where the algal movement takes on a helical trajectory. The step size of the movement is determined by the friction surface and the distance to the light source.
(18)$$X_{im}^{t+1}=X_{im}^{t}+\left(X_{jm}^{t}-X_{im}^{t}\right)\left(\Delta -\tau^{t}\left(X_{i}\right)\right)P$$
(19)$$X_{ik}^{t+1}=X_{ik}^{t}+\left(X_{jk}^{t}-X_{ik}^{t}\right)\left(\Delta -\tau^{t}\left(X_{i}\right)\right)\cos\alpha$$
(20)$$X_{il}^{t+1}=X_{il}^{t}+\left(X_{jl}^{t}-X_{il}^{t}\right)\left(\Delta -\tau^{t}\left(X_{i}\right)\right)\sin\beta$$
where $X_{im}^{t+1},X_{ik}^{t+1},X_{il}^{t+1}$ represents the x, y, and z coordinates of the ith algal cell at time *t*.

The variables *α* and *β* are in the range [0, 2π], while *p* is within the interval [−1, 1]. Δ represents the shear force and $\tau^{t}\left(X_{i}\right)$ denotes the surface area of the ith algal cell.

### 5.2. Statistical Analysis

The microarray gene data were reduced in dimension through four distinct dimensionality reduction (DR) techniques and comprehensive analysis using the statistical metrics of mean, variance, skewness, kurtosis, PCC, and CCA. This scrutiny aimed to ascertain whether the outcomes accurately portrayed the inherent properties of microarray genes within the reduced subspace. As shown in Table 2, the DR method based on AAA exhibited elevated mean and variance values across classes. In contrast, the remaining three DR methods—namely the Bessel function, Discrete Cosine Transform (DCT), and Least Squares Linear Regression (LSLR)—revealed modest and overlapping mean and variance values within classes. Among these methods, the LSLR DR approach showcased negative skewness, indicating the occurrence of skewed elements in the classes. Additionally, the DCT and LSLR DR methods demonstrated negative kurtosis, signifying their preservation of the underlying microarray gene traits. The PCC values revealed substantial correlations within the obtained outputs for a particular class. In the case of the Bessel function DR method, all four statistical parameters exhibited positive values at their minimum. This indicates an association with non-Gaussian and nonlinear distributions, a conclusion substantiated by the histograms, normal probability plots, and scatter plots of the DR method outputs. Canonical Correlation Analysis (CCA) provided insight into the correlation between DR outputs for diabetic and non-diabetic instances. Notably, the low CCA value in Table 2 suggests a limited correlation between the DR outputs of the two distinct classes.

Figure 3 shows a histogram of the Bessel function DR techniques in the diabetic class. The histogram depicts that a skewed group of values, a gap, and the existence of nonlinearity were witnessed in this method. Patients from 1 to 10 are represented as x(:,1) to x(:,10).

Figure 4 exhibits a histogram of the BF DR techniques in the non-diabetic class, in which the marker of x(:,1) represents patient 1 and x(:,10) represents patient 10. Figure 4 shows a skewed group of values, a gap, and the existence of nonlinearity in this method.

In Figure 5, data points 1 to 5 signify the reference points, 6 to 10 highlight the upper bound, and 11 to 15 depict the clustered variable points. This representation signifies the generation of a normal probability plot for features obtained using DCT DR techniques within the diabetic gene class. As can be observed from Figure 5, the plot effectively showcases the complete cluster of DCT DR outputs, accentuating the existence of variables with the nature of nonlinearity across classes.

Figure 6 shows the normal probability plot for the DCT DR techniques for the non-diabetic gene class. The data points from 1 to 5 represent references, the upper bound values are represented from 6 to 10, and the cluster variable points are from 11 to 15. The plot shows that the total cluster of DCT DR outputs and nonlinearly correlated variables among the classes were observed due to the low values of mean and variance and the presence of negative kurtosis variables in the DR method.

Figure 7 presents data points 1 to 5 as references, 6 to 10 as upper bound values, and 11 to 15 as variable points. The normal probability plot distinctly exhibits clustered groups corresponding to LSLR DR outputs. This observation underscores the existence of non-Gaussian and nonlinearly varying variables among the classes. This phenomenon can be attributed to the low variance and negative kurtosis attributes of the outcomes generated by the DR method.

Figure 8 presents the normal probability plot for LSLR DR techniques in the non-diabetic class. The plot displays a discrete group of clusters for LSLR DR outputs. The data points 1 to 5 represent references, 6 to 10 represent upper bound values, and 11 to 15 represent variable points. The flat kurtosis variable and low variance in the DR methods indicate the presence of nonlinearity and a non-Gaussian nature.

Figure 9 presents a scatter plot of the AAA DR techniques for the non-diabetic and diabetic gene classes. As can be seen, there is total clustering and overlapping of the variables in both classes. The non-Gaussian and nonlinear nature can also be observed from this graph. Furthermore, the AAA algorithm has a heavy computational cost on the classifier design. To reduce the burden of the classifiers, a feature selection process comprising the Elephant Herd Optimization (EHO) and Dragonfly algorithms was initiated.

## 6. Feature Selection Methods

The reduced dimensionality dataset was used for the feature selection methods. The metaheuristic algorithms of Monarch Butterfly Optimization (MBO) [28], Slime Mold Algorithm (SMA) [29], Moth Search Algorithm (MSA) [30], Hunger Games Search (HGS) [31], Runge Kutta Method (RUN) [32], Colony Predation Algorithm (CPA) [33], weIghtedmeaNoFvectOrs (INFO) [34], Harris Hawks Optimization (HHO) [35], Rime Optimization Algorithm (RIME) [36], Elephant Herding Optimization (EHO) [37] algorithm, and Dragonfly Optimization Algorithm (DOA) [38] were considered for the FS.

MBO has two operators: migration and butterfly adjusting operator. The Lévy flight is used in the butterfly adjusting operator, which has infinite mean and variance. SMA is used for attaining global optimization. It has three stages: the first is to make a better solution approach based on the slime mold bound condition through the iterations attained from the tanh function; the second is wrap food, based on SMA, that imitates the updating position of the slime mold; and the third is an oscillator, based on step size, which is considered within bound. MSA was also used to find the global optimization. Moths have the propensity to follow Lévy flights. It exhibits similar characteristics to MBO such as being non-Gaussian and having infinite mean and infinite variance. HGS is a good population-based optimizer; however, when dealing with challenging optimization problems, the classic HGS sometimes shows premature convergence and stagnation shortcomings. Therefore, finding approaches that enhance solution diversity and exploitation capabilities is crucial. RUN is also an optimization technique. Although RUN has a solid mathematical theoretical foundation, there are still some performance defects when dealing with complex optimization problems. In the initialization phase, the focus is on constructing a population that evolves over several iterations. CPA has taken inspiration from the predatory habits of groups in nature. However, CPA suffers from poor exploratory ability and cannot always escape certain solutions. Two strategies are used in the pursuit process to increase the probability of successful predation: scattering prey and surrounding prey. Prey dispersal drives the prey in different directions and weakens the prey group. The weIghtedmeaNoFvectOrs (INFO) algorithm is also a population-based optimization algorithm operating based on the calculation of the weighted mean for a set of vectors. It has three techniques to update the vectors’ location: a local search, a vector-combining rule, and the weighted mean concept for a solid structure. The INFO algorithm’s reliance on weighted mean vectors may not capture nonlinear relationships between features and target variables effectively. It focuses on selecting individual features based on their weighted mean values, so may not effectively explore interactions or combinations of features. HHO is a computational intelligence tool, and its complexity may increase with the number of features in high-dimensional datasets. It may struggle to handle large feature spaces efficiently, leading to longer execution times. It replicates Harris hawk predator–prey dynamics. It is divided into three sections: exploring, transformation, and exploitation. It has a high convergence rate and a powerful global search capability, but it has an unsatisfactory optimization effect on high-dimensional or complex problems. RIME is also a good optimization algorithm for search space mechanisms and the typical idea is to compare the updated fitness value of an agent with the global optimum; if the updated value is better than the current global optimum, then the optimum fitness value is replaced, and the agent is recorded as the optimum. The advantage of such an operation is that it is simple and fast, but it does not help in the exploration and exploitation of the population and only serves as a record. However, algorithms like EHO and DOA are used as feature selection parameters for emulating the behavior observed in elephants and dragonflies for the better selection of features and offer effective approaches to address the abovementioned challenges in optimization techniques for FS.

Elephant Herding Optimization (EHO) algorithm

Wang et al. [37] introduced EHO as a metaheuristic algorithm inspired by the behavior of elephants in the African savanna. It has demonstrated effectiveness in solving optimization problems and has been successfully applied in various domains, including feature selection. In feature selection, the objective is to identify a subset of informative features from a larger set that are relevant to the target variable. EHO employs a herd of elephants to search for the optimal solution, with each elephant representing a potential solution. By combining global and local search strategies, the algorithm guides the elephants towards the best solution. The methodology of the EHO is depicted in Figure 10. EHO offers immense potential as a feature selection technique due to its ability to strike a balance between global and local searches, making it suitable for high-dimensional data. The initialization of the elephant herd involves assigning random positions to the elephants in the feature space, providing a comprehensive representation of the elephants’ positions and the overall movement of the herd.
(21)$$y_{i}^{new}=y_{i}^{old}+\propto \left(Y_{best}-Y_{i}^{old}\right)*r$$

The EHO algorithm [39] involves updating the positions of elephants within the herd. This update process considers both the old position ($y_{i}^{old}$) and the new position ($y_{i}^{new}$) of each elephant. A control parameter (*α*), which falls within the range of [0, 1], is used in conjunction with a randomly generated number (r ∈ [0, 1]) to determine the new position. Additionally, each elephant in the herd maintains a memory of its best position in the feature space. The best position is updated using the following equation, ensuring that the elephant’s memory is updated accordingly.
(22)$$Y_{best}=\beta *\;Y_{centre}$$
(23)$$Y_{centre\;}=\frac{1}{m}*\;\sum_{i=1}^{m}y_{i}$$

The algorithm includes the concept of the best position (*Y_best_*) for each elephant within the herd. This best position is determined by considering the control parameter (*β*), which falls within the range of [0, 1]. The control parameter plays a role in updating and adjusting the best position of the elephant, ensuring that it reflects the optimal solution obtained during the optimization process.

By considering both the best and worst solutions, the EHO algorithm ensures a more comprehensive exploration of the solution space, leading to improved optimization performance.
(24)$$Y_{worst}=Y_{min}+\left(Y_{max}-Y_{min}+1\right)*rand$$

Dragonfly Optimization Algorithm (DOA)

The Dragonfly Algorithm (DA) is an optimization technique based on swarm intelligence, taking inspiration from the collective behaviors of dragonflies. Introduced by Mirjalili in 2016 [38], this algorithm mimics both static and dynamic swarming behaviors observed in nature. Figure 11 shows the flowchart of DOA. During the dynamic or exploitation phase, it forms large swarms and travels in a specific direction to confuse potential threats. In the static or exploration phase, the swarms form smaller groups, moving within a limited area to hunt and attract prey [40]. The DA is guided by five fundamental principles: separation, alignment, cohesiveness, attraction, and diversion. These principles dictate the behavior of individual dragonflies and their interactions within the swarm. In the equations that follow, *K* and *Ki* denote the current position and the *i*th position of a dragonfly, respectively, while *N* represents the total number of neighboring flies.

Separation: This implies that the static phase of the algorithm focuses on preventing dragonflies from colliding with each other in their vicinity. This calculation aims to ensure the avoidance of collisions among flies.
(25)$${Se}_{j}=-\sum_{i=1}^{n}k-k_{j}$$
where ${Se}_{j}$ represents the motion of the *i*th individual aimed at maintaining separation from other dragonflies.

Alignment: This denotes the synchronization of velocities among dragonflies belonging to the same group. It is represented as
(26)$${Ag}_{j}=\frac{\sum_{i=1}^{n}{Ve}_{i}}{n}$$

This is represented by ${Ag}_{j}$, which is called the velocity of the *i*th individual.

Cohesiveness: This represents the inclination of individual flies to converge towards the center of swarms. The calculation is
(27)$${Co}_{j}=\frac{\sum_{i=1}^{n}k_{i}}{N}-k$$

Attraction: The quantification of the attraction towards the food source is characterized by
(28)$$H_{j}=K^{+}-K$$

Here, $H_{j}$ is the attraction of the food source, and $K^{+}$ represents the position of the food source.

Diversion: The diversion from the enemy is determined by the outward distance, which is calculated as
(29)$$D_{j}=K^{-}+K$$

The calculation of the outward distance determines the diversion from the enemy, and it is expressed in terms of the step vector (∆*K*) and the current position vector (*K*) are used to update the locations of artificial dragonflies within the search space. The step vector (∆*K*) can be calculated using the direction of movement of the dragonfly:(30)$${∆K}_{j}^{t+1}=\left(s{Se}_{j}+a{Ag}_{j}+c{Co}_{j}+hH_{j}+dD_{j}\right)+\omega {∆K}_{j}^{t}$$

The behavior of the dragonfly algorithm is influenced by factors such as separation weight (*s*), alignment weight (*a*), cohesion weight (*c*), attraction weight (*h*), and enemy weight (*d*). The inertia weight is represented by “*ω*”, and “*t*” represents the iteration number. 

Through the manipulation of these weights, the algorithm can attain both exploration and exploitation phases. The position of the *i*th dragonfly at *t* + 1 iterations is determined by the following equation:(31)$$K_{j}^{t+1}=K_{j}^{t}+{∆K}_{j}^{t+1}$$

The evaluation of this method’s outcomes is conducted by assessing the consequence of the *p*-value using the *t*-test. Table 3 demonstrates the significance of the *p*-values associated with the EHO and Dragonfly Algorithm methods across the four DR techniques. The data presented in Table 3 reveal that both the EHO and Dragonfly Algorithms’ feature selection methods do not exhibit significant *p*-values across classes for all four dimensionality reduction methods. This *p*-value serves as an initial indicator to quantify the existence of outliers, nonlinearity, and non-Gaussian nature among the classes after the implementation of feature selection techniques.

## 7. Classification Techniques

NLR—Nonlinear Regression

The behavior of a system is expressed through mathematical equations to facilitate representation and analysis, ultimately aiming to determine an exact best-fit line between classifier values. Nonlinear regression introduces nonlinear and random variables (a, b) to capture the complexity of the system. The primary objective of nonlinear regression is to reduce the sum of squares. This involves measuring values from the dataset and computing the difference between the mean and each data point, squaring these differences, and summing them. The minimum value of the sum of squared differences indicates a better fit to the dataset.

Nonlinear models require more attention due to their inherent complexity, and researchers have devised various methods to mitigate this difficulty, such as the Levenberg–Marquardt and Gauss–Newton methods. Estimating parameters for nonlinear systems is achieved through least squares methods, aiming to minimize the residual sum of squares. Iterative techniques, including the Taylor series, steepest descent method, and Levenberg–Marquardt method (Zhang et al. [41]), can be employed for nonlinear equations. The Levenberg–Marquardt technique is commonly used for assessing the nonlinear least squares, offering advantages and producing reliable results through an iterative process.

The authors assume a represented model:(32)$$z_{i}=f\left(x_{i},\;\theta \right)+\varepsilon_{i},\;where\;i=1,\;2,\;3,\;\ldots ,\;n$$

Here, $x_{i}\;\mathrm{and}\;z_{i}$ represent the individual and supported variables of the *i*th iteration, $\theta =(\theta_{1},\theta_{2},\ldots ,\theta_{m})$ are the parameters, and $\varepsilon_{i}$ are the error terms that follow $N(0,\sigma^{2}$).
(33)$$S_{u}\left(\theta \right)=\sum_{i=1}^{n}{\left[z_{i}-f\left(x_{i},\;\theta \right)\right]}^{2}$$

Let $\theta_{k}=\theta_{1k},\theta_{2k,\;}\ldots ,\theta_{pk}$ be the starting values; the successive estimates are obtained using
(34)$$\left(H+\tau I\right)\left(\theta_{0}-\theta_{1}\right)=g$$
where $\left.g=\frac{\partial S_{u}(\theta )}{\partial \theta }\right|\theta =\theta_{o}$ and $\left.H=\frac{\partial^{2}S_{u}(\theta )}{{\partial \theta \partial \theta }^{\prime }}\right|\theta =\theta_{1},\tau$ are a multiplier and I is the identitymatrix. 

The integrity of the model is assessed using the MSE, which quantifies the discrepancy between the experimental and estimated values. The MSE is computed as the average squared difference between the actual and predicted values. The overall experimental values are expressed in terms of *N*.
(35)$$\mathrm{MSE}=\frac{1}{N}\sum_{\left(i=1\right)}^{N}({y_{i}-y_{i}^{ᴧ})}^{2}$$

The steps for nonlinear regression are the initialization of the initial parameters and the generation of curves based on these values. The goal is to iteratively modify the parameters to minimize the MSE and bring the curve closer to the desired value. The process continues until the MSE value no longer changes compared to the previous iteration, indicating convergence.

Linear Regression (LR)

In the investigation of gene expression data, linear regression is a suitable method for obtaining the best-fit curve, as the conveyance levels in the genes exhibit only minor variations. To identify the most informative genes, a feature selection process is performed by comparing the training dataset with the gene expression data within different levels of diversity. In this linear regression model, the dependent variable, denoted as *x*, is associated with the independent variable, *y*. The model aims [42] to predict values using the *x* variable, optimizing the regression fitness value based on the population in the *y* variable. The hypothesis function for a single variable is given by
(36)$$g_{\theta }=\theta_{0}+\theta_{1}x$$
where $\theta_{i}$ represents the parameters. The objective is to select the range of $\theta_{o}\;\mathrm{and}\;\theta_{1}$ that ensures $g_{\theta }$ closely approximates *y* in the training dataset (*x*, *y*).
(37)$$R\;\left(\theta_{0},\;\theta_{1}\right)=\frac{1}{2m}\sum_{i=1}^{m}{\left(g_{\theta }\left(x^{i}\right)-y^{i}\right)}^{2}$$

Here, “*m*” symbolizes the total count of samples within the training dataset. For LR models with *n* variables, the hypothesis function becomes
(38)$$g_{\theta }=\theta_{0}x_{0}+\theta_{1}x_{1}+\ldots +\theta_{n}x_{n}$$ and the cost function is given by
(39)$$R\;\left(\theta \right)=\frac{1}{2m}\sum_{i=1}^{m}{\left(g_{\theta }\left(x^{i}\right)-y^{i}\right)}^{2}$$
where *θ* is a set of parameters {$\theta_{0},\;\theta_{1},\;\theta_{2},\ldots ,\theta_{n}$}. The gradient descent algorithm is employed to minimize the cost function, and the partial derivative of the cost function is computed as
(40)$$\frac{\delta }{{\delta \theta }_{j}}\;g\left(\theta \right)=\frac{\delta }{{\delta \theta }_{j}}\sum_{i=1}^{m}{\left(g_{\theta }\left(x^{i}\right)-y^{i}\right)}^{2}$$

To update the parameter value $\theta_{j}$, the following equation is used:(41)$$\theta_{j\left(new\right)}=\theta_{j\left(old\right)}+\beta \;\frac{1}{m}\sum_{i=1}^{m}{(g}_{\theta }(x^{i}-y^{i})x_{j}^{i})$$
where *β* represents the learning rate, and $\theta_{j}$ is continuously computed until convergence is reached. In this study, *β* is set to 0.01.

The algorithm for LR involves the following steps:

Feature selection parameters, obtained from algorithms such as the Bessel function, DCT, LSLR, and AAA, are used as input for the classifiers.

A line represented by $g_{\theta }=\theta_{0}+\theta_{1}x$ is fitted to the data in a linear manner.

The cost function is formulated with the aim of minimizing the squared error existing between the observed data and the predictions.

The solutions are found by equating the derivatives of $\theta_{0}\;\mathrm{and}{\;\theta }_{1}$ to zero.

To yield the coefficient of MSE, repeat steps 2, 3, and 4. 

Gaussian Mixture Model (GMM)

GMM is a well-known unsupervised learning technique in machine learning used for many applications like pattern recognition and signal classifications. It involves integrating related objects based on clustering techniques. By classifying the data, GMM [43] facilitates the prediction and computation of unrated items within the same category. Hard and soft clustering techniques are used by GMM, and it utilizes the distribution for data analysis. Each GMM consists of multiple Gaussian distributions (referred to as “*g*”). The PDF of GMM combines these distributed components linearly, enabling easier analysis of the generated data. When generating random values as a vector “*a*” within an n-dimensional sample space χ, if “*a*” adheres to a Gaussian distribution, the expression for its probability distribution function is as follows:(42)$$p\left(a\right)=\frac{1}{{\left(2\pi \right)}^{n/2}{\left|\Sigma \right|}^{1/2}}e^{-(\frac{1}{2}){\left(a-\mu \right)}^{T}\Sigma^{-1}(a-\mu )}$$

Here, μ represents the mean vector in the n-dimensional space, and Σ is the covariance matrix of size n×n. The determination of the covariance matrix and mean vector is essential for the Gaussian distribution. Multiple components are mixed in the Gaussian distribution function [44], with each component and the equation of mixture distribution given by
(43)$$P_{Q}\left(a\right)=\sum_{i=1}^{k}\propto_{j}\times p\left(a|\mu_{j},\Sigma_{j}\right)$$

In this equation, $\propto_{j}$ represents the mixing coefficient corresponding to the jth Gaussian mixture, while $\mu_{j}\;\mathrm{and}\;\Sigma_{j}$ denote the mean vector and covariance matrix of that Gaussian component, respectively.

Expectation Maximum (EM)

The EM algorithm [45] serves as a classifier in this context. Its primary objective is to estimate missing values within a dataset and subsequently predict those values to maximize the dataset’s order based on the application’s requirements. Consider two random variables, *X* and *Y*, involved in the prediction process and determining the order of the data in rows. Variable *X* is observable and known in the dataset, while the unknown variable *Z* needs to be predicted to set the value of *Y*.
(44)$$L\;\left(\theta ;X,Y\right)=p\left(X,Y|\theta \right)$$
(45)$$L\left(\theta |x\right)\;\varepsilon \;\left\{\;\alpha p\;\left(X|\theta \right);\;\alpha >0\right\}.\;p\left(X|\theta \right)*p\left(Y|\theta \right)$$

The maximum likelihood estimation is obtained as
(46)$$L\left(\theta ;X\right)=p\;\left(X|\theta \right)=\sum_{Y}p\left(X,Z|\theta \right)$$

To estimate the expected value of the log-likelihood function, we calculate
(47)Qθθ^(t)=Ez|xi θ^(t) [log⁡ L (θ;X,Y)]

The above quantity is maximized to compute the maximum value, resulting in
(48)θ^(t+1)= arg max Qθθ^(t)

The expectation and maximization steps are iteratively repeated until a converged sequence of values is reached as mentioned in Figure 12 which is the flow diagram of expectation maximization. 

Bayesian Linear Discriminant Classifier (BLDC)

BLDC [46] is commonly employed to regularize high-dimensional signals, reduce noisy signals, and improve computational efficiency. Before conducting Bayesian linear discriminant analysis, an assumption is made that a target, denoted as *b*, is related to a vector *x* with the addition of white Gaussian noise, *c*.

This relationship can be expressed as $a=x^{T}b+c$. The function *x* is assigned weights, and its likelihood function is given by
(49)$$p\left(G|\beta ,x\right)={\left(\frac{\beta }{2\pi }\right)}^{\frac{c}{2}}\exp\left(-\frac{\beta }{2}\left\|B^{T}x-m\right\|\right)$$
where the pair $\left\{B,m\right\}$ represents *G*. The x of prior distribution is expressed as
(50)$$p\left(x|\alpha \right)={\left(\frac{\alpha }{2\pi }\right)}^{\frac{1}{2}}{\left(\frac{\varepsilon }{2\pi }\right)}^{\frac{1}{2}}\exp\left(-\frac{1}{2}x^{T}H^{\prime }\left(\alpha \right)x\right)$$

The regularization square matrix is given by
(51)$$H^{\prime }(\alpha )={\left[\begin{array}{ccc} \alpha & \cdots & 0\\ \vdots & \ddots & \vdots \\ 0 & \cdots & \varepsilon \end{array}\right]}_{\left(l+1)(l+1\right)}$$ and α is a hyperparameter obtained from data forecasting, while *l* represents the assigned vector number. By applying Bayes’ rule, x can be calculated as
(52)$$p\left(x|\beta ,\alpha ,G\right)=\frac{P\left(G|\beta ,x\right)P\left(x|\alpha \right)}{\int P\left(G|\beta ,x\right)P\left(y|\alpha \right)dy}$$

The mean vector υ and the covariance matrix X must adhere to the specific norms outlined in Equations (50) and (52) for the posterior distribution. The predominant nature of the posterior distribution is Gaussian.
(53)$$\upsilon =\beta {\left(\beta BB^{T}+H^{\prime }\left(\alpha \right)\right)}^{-1}Ba$$
(54)$$X={\left(\beta BB^{T}+H^{\prime }\left(\alpha \right)\right)}^{-1}$$

When predicting the input vector $\hat{b}$, the probability distribution for regression can be expressed as
(55)$$p\left(\hat{a}|\beta ,\alpha ,\hat{b,G}\right)=\int p\left(\hat{a}|\beta ,\hat{b,x}\right)p\left(x|\beta ,\alpha ,G\right)dy$$

Again, the nature of this prediction analysis is predominantly Gaussian, with the mean expressed as $\mu =\upsilon^{T}\hat{b}$ and variance expressed as $\delta^{2}=\frac{1}{\beta }+\hat{b^{T}}X\hat{b}$.

Logistic Regression (LoR)

Logistic Regression (LoR) has proven to be effective in classifying diseases such as diabetes, types of cancer, and epilepsy. In this context, function *y* that represents the disease level is considered, ranging from 0 to 1 to indicate non-diabetic and diabetic patients, respectively. Gene expressions are represented by a vector $x=x_{1,\;}x_{2},\;\ldots ,\;x_{m}$, where each element $x_{j}$ corresponds to the expression level of the *j*th gene. Using a model-based approach for a dataset Π(x), the aim is to identify informative genes for diabetic patients based on the likelihood of *y* being 1 given *x*. To achieve dimensionality reduction, logistic regression is utilized to select the most relevant “*q*” genes. The gene expression representation $x_{j}$^*^ corresponds to the gene expression, with *j* ranging from 1 to *q*, while the binary disease status is denoted by $y_{i}$, where *i* ranges from 1 to *n*. The logistic regression model can be expressed as
(56)$$Logit\left\{\Pi \left(x\right)\right\}=\upsilon_{0}+\sum_{j=1}^{q}\upsilon_{j}x_{j}^{*}$$

The objective is to maximize the fitness and log-likelihood, which can be achieved by obtaining the following function
(57)$$1\left(\upsilon_{0},\upsilon \right)=\sum_{j=1}^{n}\left\{y_{i}\log\left(\pi_{i}\right)+\left(1-\pi_{i}\right)\right\}-\frac{1}{{2\tau }^{2}}{\left\|\upsilon \right\|}^{2}$$
where *τ* is a parameter that limits the reduction in *υ* near 0, $\pi_{i}=\pi (x_{i})$ as defined by the model [47,48], and ${\left\|\upsilon \right\|}^{2}$ denotes the Euclidean length of $\upsilon =\upsilon_{1},\upsilon_{2},\ldots ,\upsilon_{p}$. The selection of q and τ are determined using the parametric bootstrap method, which imposes constraints on accurate error prediction. Initially, υ = 0, for the purpose of calculating the cost function. It is then varied with different parameters to minimize the cost function. The sigmoid function is applied to restrict values between 0 and 1, serving as an attenuation mechanism. The threshold cut-off value of 0.5 is used to classify patients as either diabetic or non-diabetic. Any probability below the threshold is considered indicative of non-diabetic patients, while values above the threshold indicate diabetic patients.

SDC—Softmax Discriminant Classifier

The SDC is used to verify and detect the group to which a particular test sample belongs [49]. It weighs the distance between the training samples and the test sample within a particular class or group of data. Z is represented as
(58)$$Z=\left[Z_{1,}Z_{2},\ldots ,Z_{q}\right]\in R^{c\times d}$$
consisting of samples from distinct classes named *q*, $Z_{q}=\left[Z_{1}^{q},Z_{2}^{q},\;\ldots ,Z_{dq}^{q}\right]\in \;R^{c\times d_{q}}$. Each class, represented by $Z_{q}$, contains samples from the qth class, where $\sum_{i=1}^{q}d_{i}=d.\;$ The sum of the sample sizes, given a test sample $K\;\in \;R^{c\times 1}$, is passed through the classifiers to obtain minimal construction errors, thereby assigning it to the class *q*. The transformation of class samples and test samples in SDC involves nonlinear enhancement values. This is achieved through the following equations:(59)$$h\left(K\right)=\arg\max Z_{w}^{i}$$
(60)$$h\left(K\right)=\arg\underset{i}{\max}\log\left(\sum_{j=1}^{d_{i}}\exp\left(-\lambda {\left\|v-\upsilon_{j}^{i}\right\|}_{2}\right)\right)$$

In these equations, $h\left(K\right)$ represents the differentiate of the ith class and ${\left\|v-\upsilon_{j}^{i}\right\|}_{2}$ approaches zero, resulting in the maximization of $Z_{w}^{i}$. This asymptotic behavior leads to the maximum likelihood of the test sample belonging to a particular class.

Support Vector Machines

The SVM classifier is a significant machine learning approach widely used for classification problems, particularly in the phase of nonlinear regression [50]. In this study, three distinct methods are explored for data classification:

SVM-Linear: this method utilizes a linear kernel to classify the data.

SVM-Polynomial: this approach involves the use of a polynomial kernel for data classification.

SVM-Radial Basis Function (RBF): the RBF kernel is used here to classify the data.

These three SVM methods offer different strategies for effectively classifying datasets, allowing researchers to choose the most suitable approach based on their specific classification requirements.

The training time and computational complexity of the SVM depend on the data and classifiers used. When the number of supports in the SVM increases, it results in higher computational requirements due to the calculation of floating-point multiplications and additions. To address this issue, K-means clustering techniques have been introduced to reduce the number of supports in the SVM. In the linear case, Lagrange multipliers can be employed, and the data points on the borders are expressed as $\nu =\sum_{i=1}^{m}\alpha_{i}z_{i}y_{i}^{T}$. Here, *m* represents the number of supports, $z_{i}$ represents the target labels for y, and the linear discriminant function is used.
(61)$$h\left(y\right)=sgn\left(\sum_{i=1}^{m}\alpha_{i}z_{i}y_{i}^{T}y+C\right)$$

The process of implementing the Support Vector Machine (SVM) involves several key steps.

*Step 1*: The first step is to use quadratic optimization to linearize and converge the problem. By transforming the primal minimization problem into a dual optimization problem, the objective is to maximize the dual Lagrangian *L_D_* with respect to $\alpha_{i}$:(62)$$MaxL_{D}=\sum_{i=1}^{l}\alpha_{i}-\frac{1}{2}\sum_{i=1}^{l}\sum_{j=1}^{l}\alpha_{i}\alpha_{j}y_{i}y_{j}(X_{i}\cdot X_{j})$$
subject to $\sum_{i=1}^{l}\alpha_{i}y_{i}=0$, where $\alpha_{i}\geq 0\forall i=1,\;2,\;3,\;\ldots ,\;l$.

*Step 2*: The next step involves solving the quadratic polynomial programming to obtain the optimal separating hyperplane. The data points with non-zero Lagrangian multipliers ($\propto_{i}>0$) are identified as the support vectors.

*Step 3*: The optimal hyperplane is determined based on the support vectors, which are the data points closest to the decision boundary in the trained data.

*Step 4*: K-means clustering is applied to the dataset, grouping the data into clusters according to the conditions from Steps 2 and 3. Three points are randomly chosen from each cluster as the center points, which are representative points from the dataset. Each center point acquires the points around them.

*Step 5*: When there are six central points, each representing an individual cluster, the SVM training data are acquired through the utilization of kernel methods.
Polynomial Function: K (X, Z) = (X^T^ Z + 1) ^d^(63)(64)$$\mathrm{Radial}\;\mathrm{Basis}\;\mathrm{Function}:\;k\left(x_{i},x_{j}\right)=\exp\left\{\frac{-{\left|x_{i}-x_{j}\right|}^{2}}{{\left(2*\sigma \right)}^{2}}\right\}$$

### 7.1. Training and Testing of Classifiers

Due to the limited availability of training data, we employed k-fold cross-validation, a widely used technique for evaluating machine learning models. The methodology described by Fushiki et al. [51] was followed to conduct the k-fold cross-validation. Initially, the dataset was divided into k equally sized subsets or “folds”. This process was repeated for all k-folds, ensuring that each fold was used once for testing. Consequently, k performance estimates (one for each fold) were obtained. To obtain an overall estimate of the model’s performance, the average of these k performance estimates was calculated. After training and validating the model using k-fold cross-validation, it was retrained on the complete dataset to make predictions on new, unseen data. The significant advantage of utilizing this method is the more reliable model performance compared to other test split methods, as the technique maximizes the utilization of the available data. Here, we adopted a k-value of 10-fold cross-validation. Furthermore, the research incorporated 2870 dimensionally reduced features per patient, focusing on a cohort of 20 patients with diabetes and 50 non-diabetic patients. The utilization of cross-validation eliminates any reliance on a specific pattern for the test set, enhancing the robustness of our findings. The training process is regulated in the MSE proposed by Wang et al. [52], which is defined as follows:(65)$$\mathrm{MSE}=\frac{1}{N}\sum_{j=1}^{N}{\left(O_{j}-T_{j}\right)}^{2}$$
where *O_j_* is the observed value at time *j*, and *T*_*j*_ is the target value at model *j*.

Table 4 represents confusion matrix for detecting diabetes. The following terms in Table 4 can be defined as:

TP—true positive: a patient is accurately classified into the diabetic class.

TN—true negative: a patient is accurately recognized as belonging to the non-diabetic class.

FP—false positive: a patient is inaccurately classified as belonging to the diabetic class when they actually belong to the non-diabetic class.

FN—false negative: a patient is inaccurately classified as being in the non-diabetic class when they should be categorized as belonging to the diabetic class.

**Table 4 biomimetics-08-00503-t004:** Confusion matrix for detecting diabetes.

Clinical Situation		Predicted Values	
		Diabetic	Non-Diabetic
Real Values	Diabetic class	TP	FN
	Non-diabetic class	FP	TN

Table 5 provides insight into the performance of the classifiers without the feature selection method, focusing on the training and testing Mean Squared Error (MSE) across various DR techniques. The training MSE values consistently range between 10^−4^ and 10^−10^, while the testing MSE varies from 10^−4^ to 10^−8^. Among the classifiers, the SVM (RBF) classifier using the AAA DR technique without feature selection achieves the lowest training and testing MSE, specifically 1.93 × 10^−10^ and 1.77 × 10^−8^, respectively. Notably, a lower testing MSE indicates superior classifier performance. It is evident from Table 5 that higher testing MSE values correspond to lower classifier performance, regardless of the DR techniques used.

Table 6 exhibits the training and testing of MSE in the classifiers with EHO feature selection method across all four DR techniques. The training MSE varies from 10^−5^ to 10^−10^, while the testing MSE varies between 10^−5^ and 10^−8^. The SVM (RBF) classifier in the AAA DR method with PSO feature selection achieved a minimum training and testing MSE of 1.99 × 10^−10^ and **2.5** × 10^−8^, respectively. The Bessel function DR method indicates slightly lower training and testing MSE values for the classifiers when compared to the other three DR techniques. All of the classifiers had slightly enhanced testing performance when compared to methods without feature selection. This indicates the enhancement in classifier performance irrespective of the DR technique.

Table 7 demonstrates the training and testing Mean Squared Error (MSE) performance of classifiers utilizing the Dragonfly Algorithm-based feature selection method across various dimensionality reduction techniques. The training MSE values range from 10^−6^ to 10^−9^, while the testing MSE varies between 10^−5^ and 10^−8^. The SVM (RBF) classifier, when combined with the Dragonfly feature selection method, achieved a minimal training MSE of 1.66 × 10^−9^ and a testing MSE of **3.25** × 10^−8^. Notably, this feature selection method led to improvements in the training and testing performance of all classifiers. This enhancement is reflected in improved accuracy, MCC, and Kappa parameters, regardless of the specific dimensionality reduction technique employed.

### 7.2. Selection of Target

The non-diabetic class ${(T}_{ND})$ target value is taken at the lower side from 0→1, and this is mapped according to the following constraint:(66)$$\frac{1}{N}\sum_{i=1}^{N}\mu_{i}\leq T_{ND}$$

Here, $\mu_{i}$ represents the mean value of the input feature vectors for the *N* number of non-diabetic features considered for classification. Similarly, for the diabetic class ($T_{Dia}$)), the target value is mapped to the upper end of the zero-to-one (0→1) scale. This mapping is established based on the following:(67)$$\frac{1}{M}\sum_{j=1}^{N}\mu_{j}\leq T_{Dia}$$

Here, $\mu_{j}$ signifies the average value of input feature vectors for the *M* number of diabetic cases used for classification. It is important to highlight that the target value $T_{Dia}$ is set to be higher than the average values of $\mu_{i}$ and $\mu_{j}$. This selection of target values requires the discrepancy between them to be at least 0.5, as expressed by the following:(68)$$||T_{Dia}-T_{ND}||\geq 0.5$$

The targets for non-diabetic $T_{ND}$ and diabetic $T_{Dia}$ are chosen as 0.1 and 0.85, respectively. Once the target is fixed, MSE is used for evaluating the performance of the classifiers. Table 8 shows the selection of optimal parameters for the classifiers after training and testing process. 

## 8. Results and Findings

The study employs the conventional tenfold testing and training approach, where 10% of the input is dedicated to testing, while the remaining 90% is utilized for training. The selection of performance metrics is pivotal for assessing the efficacy of classifiers. The assessment of classifier performance, especially in binary classification scenarios like distinguishing between diabetic and non-diabetic cases from pancreatic microarray gene data, relies on the utilization of a confusion matrix. This matrix facilitates the computation of performance metrics including accuracy, F1 score, MCC, error rate, FM metrics, and Kappa, which are commonly utilized to gauge the comprehensive performance of the model. The relevant parameters associated with the classifiers for performance analysis are illustrated in Table 9.

The performance of the classifier was evaluated using several metrics, including Acc, F1 score, MCC, ER, FM, and Kappa. Accuracy is the fraction of predictions that are correct, and it is a measure of the overall performance of the classifier. F1 score is the harmonic mean of precision and recall, and it is a measure of the classifier’s ability to both correctly identify positive instances and to correctly identify negative instances. MCC is a measure of the correlation between the observed and predicted classifications, and it is a more sensitive metric than accuracy or F1 score. Error rate is the fraction of predictions that are incorrect, and it is the complement of accuracy. The FM metric is a generalization of the F-measure that adds a beta parameter, and it is a measure of the classifier’s ability to both correctly identify the values of positive and negative instances, with a weighting that can be adjusted to favor one or the other. Kappa is a statistic that measures agreement between observed and predicted classifications, adjusted for chance. The results are tabulated in Table 10.

Table 10 illustrates the performance analysis of ten classifiers, considering metrics such as Acc, F1 score, MCC, ER, F-measure, and Kappa values. This analysis is conducted for four DR methods without the incorporation of two feature selection methods. Table 9 reveals that the EM classifier in the Bessel function DR technique achieves a moderate accuracy of 61.42%, an F1 score of 54.23%, a moderate error rate of 38.57%, and an F-measure of 57.28%. However, the EM classifier exhibits a lower MCC value of 0.3092 and a Kappa value of 0.2645. On the other hand, the SVM (linear) classifier in the Bessel function DR method demonstrates a low accuracy of 52.85% along with a high error rate of 47.15%. Additionally, it exhibits an F1 score of 40% and an F-measure of 41.57%. The MCC and Kappa values for the SVM (linear) classifier are notably low, at 0.06324 and 0.05714, respectively. Across the Bessel function DR techniques, all classifiers exhibit poor performance in the various metrics. This trend can be attributed to the intrinsic properties of the Bessel function, which is evident from the non-negative values of the statistical parameters. Equally, the SVM (RBF) classifier in the context of the DCT DR technique achieves a respectable accuracy of 88.57%, complemented by a low error rate of 11.42%. Furthermore, it attains an F1 score of 81.81% and an F-measure of 82.15%. The MCC and Kappa values of the SVM (RBF) classifier reach 0.7423 and 0.7358, respectively. Within the SVM (RBF) classifier, the AAA DR technique exhibits a remarkable accuracy score of 90%, coupled with a low error rate of 10%. This is accompanied by an F1 score of 84.44% and an F-measure of 84.97%. The MCC and Kappa values of the SVM (RBF) classifier are noteworthy, totaling 0.7825 and 0.772, respectively. Remarkably, regardless of the DR technique employed, all classifiers manage to maintain accuracy within the range of 52% to 85%. This is primarily due to the inherent limitations of the DR techniques. Therefore, incorporating feature selection methods is highly recommended to enhance the performance of these classifiers.

Figure 13 provides an overview of the performance analysis of ten classifiers concerning the metrics of accuracy, F1 score, error rate, and F-measure values. This analysis is carried out within the context of four dimensionality reduction methods, specifically without feature selection methods. Table 10 shows that the EM classifier in the Bessel function DR technique achieves a modest accuracy of 61.42%, along with an F1 score of 54.23%. Moreover, it exhibits a moderate error rate of 38.57% and an F-measure of 57.28%. On the other hand, the SVM (linear) classifier in the Bessel function DR method demonstrates a lower accuracy of 52.85%. This classifier is accompanied by a higher error rate of 47.15%, an F1 score of 40%, and an F-measure of 41.57%. Across the performance metrics, all classifiers exhibit suboptimal performance within the Bessel function DR technique. This trend is observed consistently across various measures. However, the SVM (RBF) classifier within the DCT DR technique maintains an impressive accuracy level of 88.57%. Furthermore, it exhibits a commendably low error rate of 11.42%, an F1 score of 81.81%, and an F-measure of 82.15%. Employing the AAA DR technique in the SVM (RBF) classifier results in achieving an elevated accuracy rate of 90%. Additionally, this combination yields a notably low ER of 10% and an F1 score of 84.44%, accompanied by an F-measure of 84.97%.

Table 11 presents an in-depth analysis of the performance of ten classifiers concerning four DR methods integrated with the EHO feature selection technique. Notably, the SVM (RBF) classifier within the AAA DR technique achieves an exceptional accuracy of 95.71%. This classifier further demonstrates a commendable F1 score of 92.68%, accompanied by a notably low error rate of 4.28% and an impressive F-measure of 92.71%. Additionally, the SVM (RBF) has a high MCC value of 0.897 and a Kappa value of 0.8965. However, a contrasting performance is observed with the SVM(Linear) classifier within the Bessel function DR technique. Once again, this classifier registers a relatively low accuracy of 50%, coupled with a high error rate of 50%. Further metrics include an F1 score of 36.36% and an F-measure of 37.79%. Intriguingly, the SVM (Linear) classifier achieves null values for both MCC and Kappa, marking a unique and distinctive characteristic of its performance. All classifiers exhibit improved accuracy within the DCT, LSLR, and AAA DR techniques. However, the impact of the EHO feature selection method does not translate into substantial enhancements for classifiers employing the Bessel function DR method.

Figure 14 presents the analysis of the ten classifiers concerning the four DR methods combined with the EHO feature selection techniques. Furthermore, it is evident from the insights presented in Table 11 that the SVM (RBF) classifier, operating within the AAA DR technique, achieves an impressively high accuracy of 95.71%. Additionally, this classifier demonstrates a notable F1 score of 92.68%, accompanied by a commendably low error rate of 4.29% and an impressive F-measure of 92.71%. Equally, the SVM(Linear) classifier used within the Bessel function DR technique reflects a lower accuracy of 50%, coupled with a higher error rate of 50%. Correspondingly, the F1 score is registered at 36.36%, and the F-measure reaches 37.79%. Overall, the classifiers exhibit relatively low performance within the context of the Bessel function DR technique.

Table 12 presents the analysis of the ten classifiers concerning the four DR methods combined with the Dragonfly method. As depicted in Table 12, it is evident that the SVM (RBF), operating within the AAA DR technique, achieves an impressively high accuracy rate of 94.28%. Moreover, this classifier demonstrates a commendable F1 score of 90.47%, accompanied by a relatively low error rate of 5.72% and an appreciable F-measure of 90.57%. Furthermore, the SVM (RBF) classifier exhibits notable values of MCC and Kappa, standing at 0.866 and 0.864, respectively. On the other hand, the SVM (Polynomial) classifier, applied within the context of the Bessel function DR technique, achieves a lower accuracy rate of 58.57%. Correspondingly, it registers a higher error rate of 41.43%, along with an F1 score of 43.13% and an F-measure of 44.17%. However, the MCC and Kappa values for the SVM (Polynomial) classifier are notably lower, reaching 0.1364 and 0.1287, respectively. Among the classifiers utilized in the Bessel function DR method, only the SVM (RBF) classifier achieves an accuracy above 78%. Additionally, the SVM (RBF) classifier attains high accuracy in the DCT DR and LSLR DR methods, reaching 91% and 90%, respectively.

Figure 15 illustrates the performance assessment of the ten classifiers concerning the four DR methods, paired with the Dragonfly feature selection technique. It is observed from Table 12 that the SVM (RBF) classifier, within the AAA DR technique, attains a notably high accuracy rate of 94.28%. This classifier also demonstrates a commendable F1 score of 90.47%, coupled with a comparatively low ER of 5.72%, and a noteworthy F-measure of 90.57%. Conversely, the SVM (Polynomial) classifier, employed in the context of the Bessel function DR technique, registers a relatively low accuracy of 58.57%. Correspondingly, it records a higher error rate of 41.43%, accompanied by an F1 score of 43.13%, and an F-measure of 44.17%. Among the four dimensionality reduction methods, the SVM (RBF) classifier consistently achieves individual accuracy levels exceeding 81%. However, it is important to note that the classifier’s performance in the Bessel function DR method, when paired with the Dragonfly feature selection, remains in the lower performance category.

Figure 16 presents the comparative analysis of the MCC and Kappa parameters across the various classifiers concerning the four different DR techniques. This analysis was conducted for measuring the MCC and Kappa that serve as benchmarks, shedding light on the performance outcomes of the classifiers across diverse inputs. In this study, the inputs are categorized into three groups: dimensionally reduced without feature selection, with EHO feature selection, and with Dragonfly feature selection methods. These classifiers’ performance is evaluated based on the MCC and Kappa values derived from these inputs. The average MCC and Kappa values across the classifiers are calculated to be 0.2984 and 0.2849, respectively. A systematic approach is formulated to assess the classifiers’ performance, drawing insights from Figure 14. The MCC values are categorized into three ranges: 0.0–0.25, 0.251–0.54, and 0.55–0.9. Notably, the classifiers exhibit poor performance within the first range, while the MCC vs. Kappa slope demonstrates a significant upsurge within the second range of MCC values. In contrast, the third range of MCC values corresponds to a higher level of classifier performance, devoid of any substantial anomalies.

Figure 17 shows a histogram of the error rate and MCC (%) parameters that were analyzed. It can be seen that the maximum error rate is 50% and the maximum MCC is 90%. The histogram of the error rate is skewed at the right side of the graph, which indicates that for any of the DR methods, and irrespective of the feature selection method, the classifier’s error rate does not go beyond 50%. The histogram of MCC depicts the classifier as being sparser at the edges and covering more points in the middle area.

### 8.1. Computational Complexity (CC)

The analysis of the classifiers in this study considers their CC, which is determined based on the size of the input (denoted as O(n)). A lower CC, indicated by O(1), is desirable as it indicates that the complexity remains constant regardless of the input size. However, it is directly proportional to the number of inputs and computational complexity. It is notable that CC is independent of the size of the input, which is a favorable characteristic for any type of algorithm. If it increases logarithmically with the increase in ‘n’, it is represented as O(logn). Additionally, hybrid models of classifiers are used that incorporate DR techniques and feature selection methods in their classification process.

Table 13 presents the CC of the classifiers without incorporating feature selection methods. A noteworthy observation from the table is that the CC of all of the classifiers is relatively similar. However, their performance in terms of accuracy is relatively low. Among the classifiers, the Bessel function classifier demonstrates a moderate CC of O(n^3^logn), while the Discrete Cosine Transform, Least Squares Linear Regression, and Artificial Algae Algorithm exhibit higher CC with improved accuracy, represented by O(2n^4^log2n), O(2n^5^log4n), and O(2n^5^log8n), respectively, when compared to the other classifiers. Additionally, when considering the values of MCC and Kappa, the DST, LSLR, and AAA classifiers exhibit similar performance.

Table 14 illustrates the CC of the classifiers utilizing the EHO feature selection method. The table reveals that the CC of all of the classifiers is relatively similar, while their performance demonstrates significant accuracy. Similar to the case without feature selection, the Expectation Maximum classifier exhibits a higher computational complexity of O(n^5^logn) along with remarkable accuracy. Regarding the DCT, LSLR, and AAA classifiers, they achieve similar CC to the SVM (RBF) classifier with O(2n^6^log2n), O(2n^7^log4n), and O(2n^7^log8n), respectively. Notably, the SVM (RBF) classifier in combination with the EHO feature selection technique for DCT, LSLR, and AAA achieves the highest accuracy among all classifiers, with accuracies of 90%, 88.57%, and 95.71%, respectively. Furthermore, the corresponding Kappa values for these classifiers are 0.7655, 0.65, and 0.8965, indicating their strong performance.

Table 15 provides insights into the CC of the classifiers using the Dragonfly method. From the table, it can be seen that the CC of all of the classifiers is relatively similar, while their performance exhibits a significant level of accuracy. Notably, all four dimensionality reduction techniques demonstrate the highest CC compared to their counterparts. Specifically, the Bessel function, DCT, LSLR, and AAA classifiers achieve a computational complexity of O(8n^5^log2n), O(8n^5^log2n), O(8n^6^log4n), and O(8n^6^log8n), respectively. Regarding accuracy, the Bessel function, DCT, LSLR, and AAA classifiers achieve the highest accuracy values of 81.42%, 91.42%, 90%, and 95.71%, respectively. Moreover, the corresponding Kappa values for these classifiers are 0.538, 0.796, 0.772, and 0.864, indicating their robust performance. A comparison with previous work is provided in Table 16.

As observed in Table 16, it is evident that a variety of machine learning classifiers, including SVM (RBF), NB, LoR, DT, NLR, RF, multilayer perceptron, and DNN, have been employed for diabetic classification using clinical databases. The accuracies of these classifiers span the range of 67% to 95%. However, the present investigation focuses on diabetes detection using microarray gene data, where SVM (RBF) stands out with an accuracy of 95%.

### 8.2. Limitations and Major Outcomes

The findings of this study may be limited to the specific population of type II diabetes mellitus patients and may not be applicable to other populations or different types of diabetes. The analysis in this study relies on microarray gene data, which may not be readily available or accessible in all healthcare settings. The methods proposed in this study, such as microarray gene arrays, may involve complex and expensive procedures that are not feasible for routine clinical practice. The performance of the classifiers in this study may be influenced by the presence of outliers in the data. Outliers can have a significant impact on the accuracy and reliability of the classification results. The developed classification approach, which utilizes various dimensionality reduction techniques and feature selection methods, has demonstrated its potential in effectively screening and predicting diabetic markers, while also identifying associated diseases such as strokes, kidney failure, and neuropathy. An outcome of this study is the establishment of a comprehensive database for the mass screening and sequencing of diabetic genomes. By incorporating microarray gene data and leveraging the proposed classification techniques, this database enables the identification of patterns and trends in diabetes outbreaks associated with different lifestyles.

The ability to detect diabetes in its early stages and predict associated diseases is of utmost importance for chronic diabetic patients. This will facilitate timely interventions, improve disease management, and, ultimately, lead to better patient outcomes. Overall, this study contributes valuable insights to the field and lays the foundation for further investigations into the early detection and management of type II diabetes mellitus patients.

## 9. Conclusions

The results showed that the classifiers exhibited lower accuracy and other performance metrics when using the BF-DR technique, which can be attributed to the inherent limitations of the Bessel function. However, the DCT and LSLR techniques produced improved accuracy and performance metrics for specific classifiers, such as the SVM (RBF) classifier. In particular, the AAA technique, combined with the SVM (RBF) classifier, achieved the highest accuracy of 90% without feature selection. The SVM (RBF) classifier in combination with the EHO feature selection technique achieved the highest accuracy values of 81.42, 90%, 88.57%, and 95.71% for BF, DCT, LSLR, and AAA, respectively. With the use of the Dragonfly feature selection method, which also showed promising results, the classifiers achieved high accuracy values of 81.42%, 91.42%, 90%, and 94.28% for BF, DCT, LSLR, and AAA, respectively. In terms of computational complexity, we observed that the classifiers exhibited similar complexities across the different dimensionality reduction techniques. However, their performance in terms of accuracy varied significantly. Notably, the SVM (RBF) classifier in combination with the EHO feature selection technique consistently achieved the highest accuracy values across the different dimensionality reduction techniques. In conclusion, this research article presents a novel method for detecting type II DM using microarray gene data. Future work will be carried out in the direction of the Convolution Neural Network (CNN), Deep Learning Network (DNN), LSTM, and hyperparameter tuning of classifiers. Moreover, this approach will be used for continuous monitoring in clinical practice.

## Figures and Tables

**Figure 1 biomimetics-08-00503-f001:**
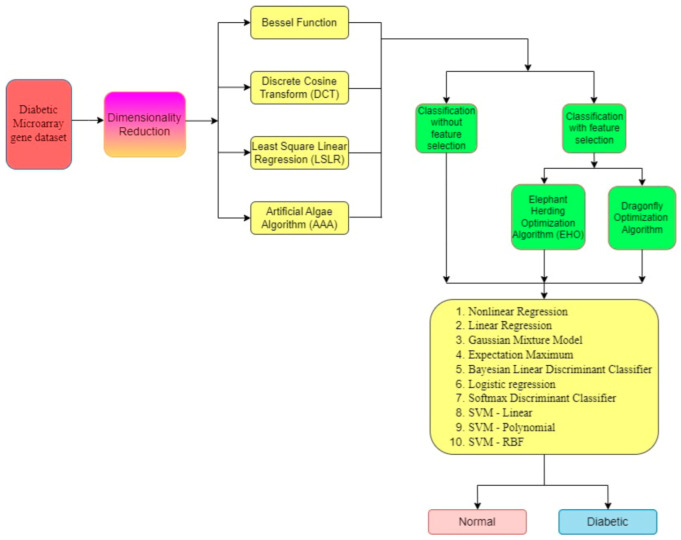
Workflow diagram.

**Figure 2 biomimetics-08-00503-f002:**
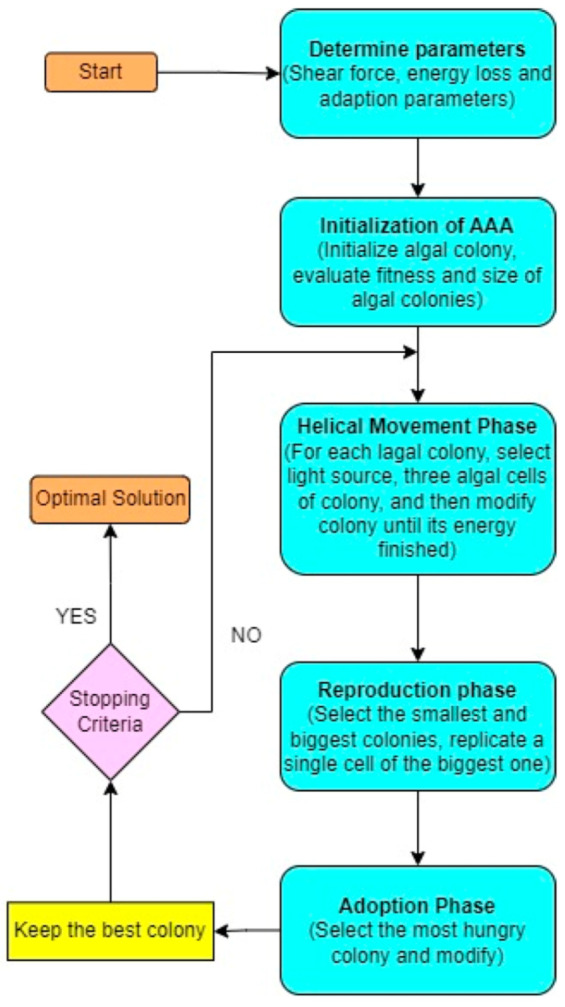
Flow diagram for Artificial Algae Algorithm.

**Figure 3 biomimetics-08-00503-f003:**
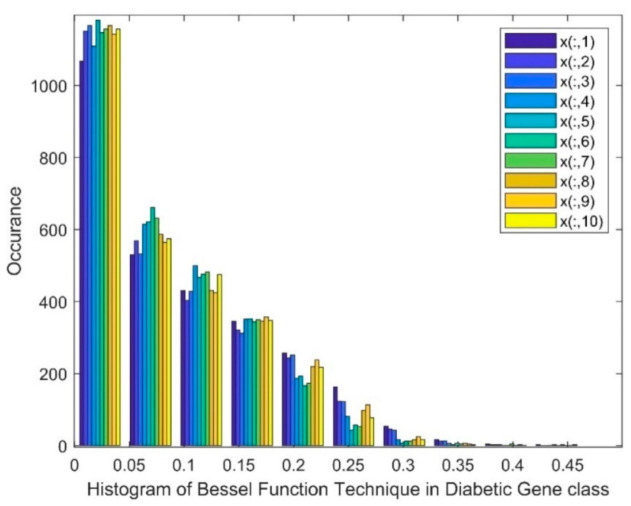
Histogram of Bessel function technique in the diabetic gene class.

**Figure 4 biomimetics-08-00503-f004:**
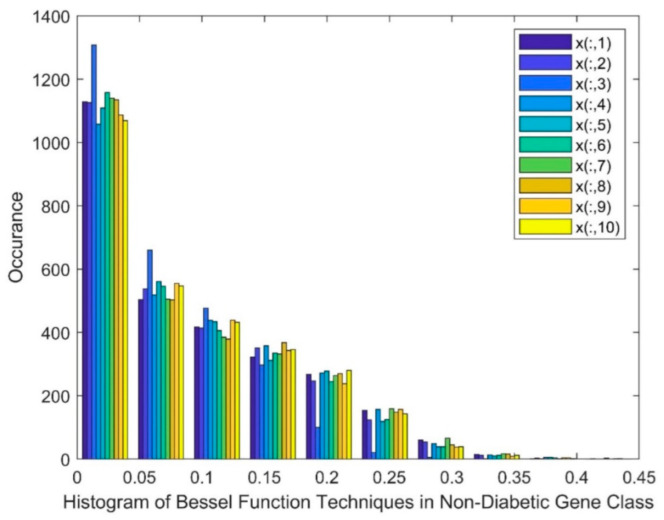
Histogram of Bessel function technique in the non-diabetic gene class.

**Figure 5 biomimetics-08-00503-f005:**
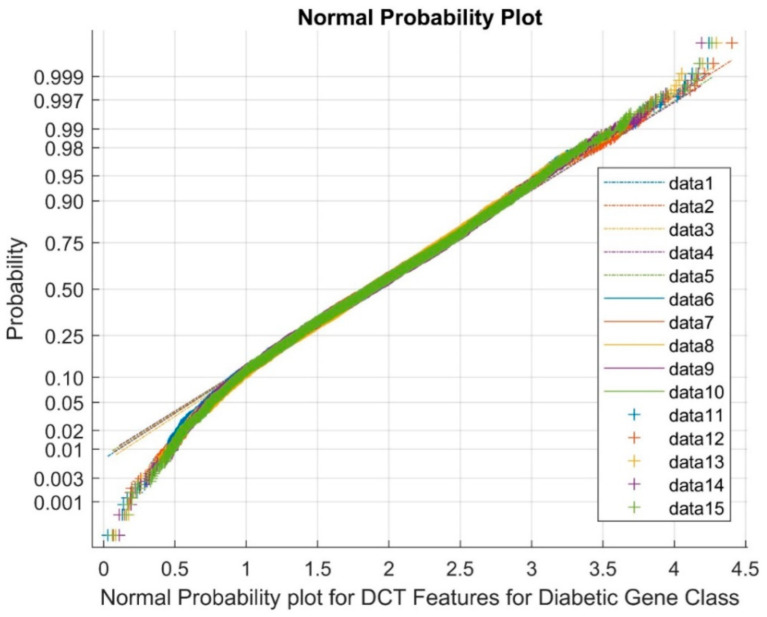
Normal probability plot showcasing DCT features for the diabetic gene class.

**Figure 6 biomimetics-08-00503-f006:**
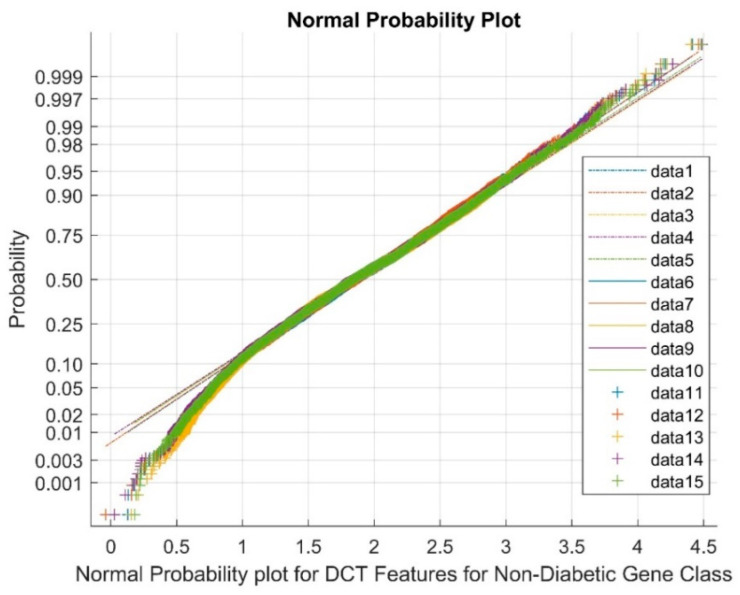
Normal probability plot representing DCT features for the non-diabetic gene class.

**Figure 7 biomimetics-08-00503-f007:**
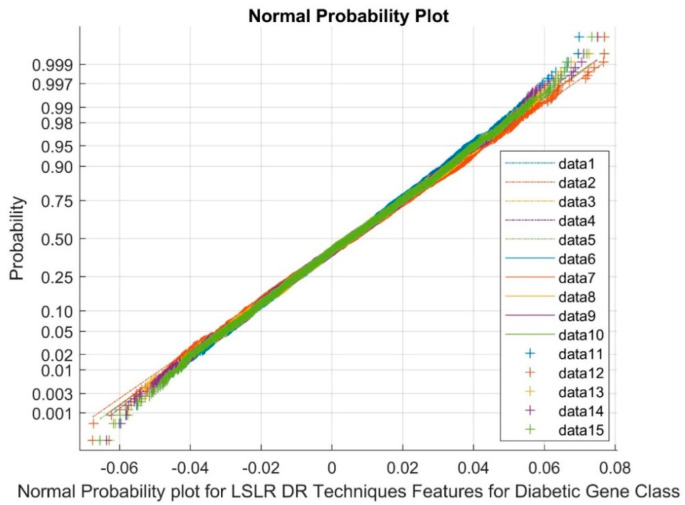
Normal probability plot for LSLR DR techniques in diabetic gene class.

**Figure 8 biomimetics-08-00503-f008:**
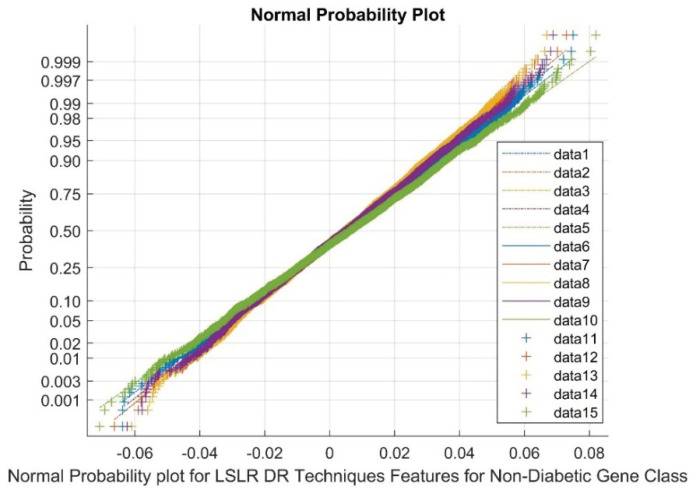
Normal probability plot for LSLR DR techniques in non-diabetic gene class.

**Figure 9 biomimetics-08-00503-f009:**
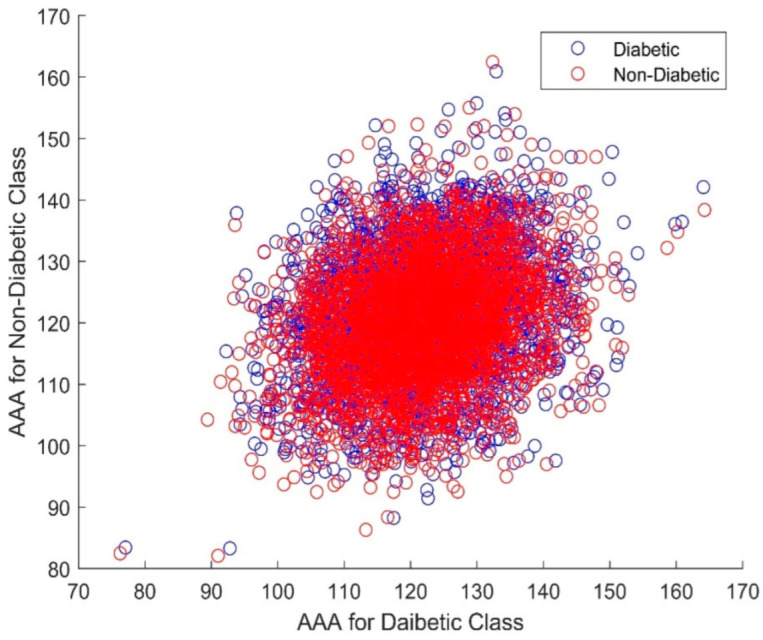
Scatter plot depicting AAA DR results for both non-diabetic and diabetic classes.

**Figure 10 biomimetics-08-00503-f010:**
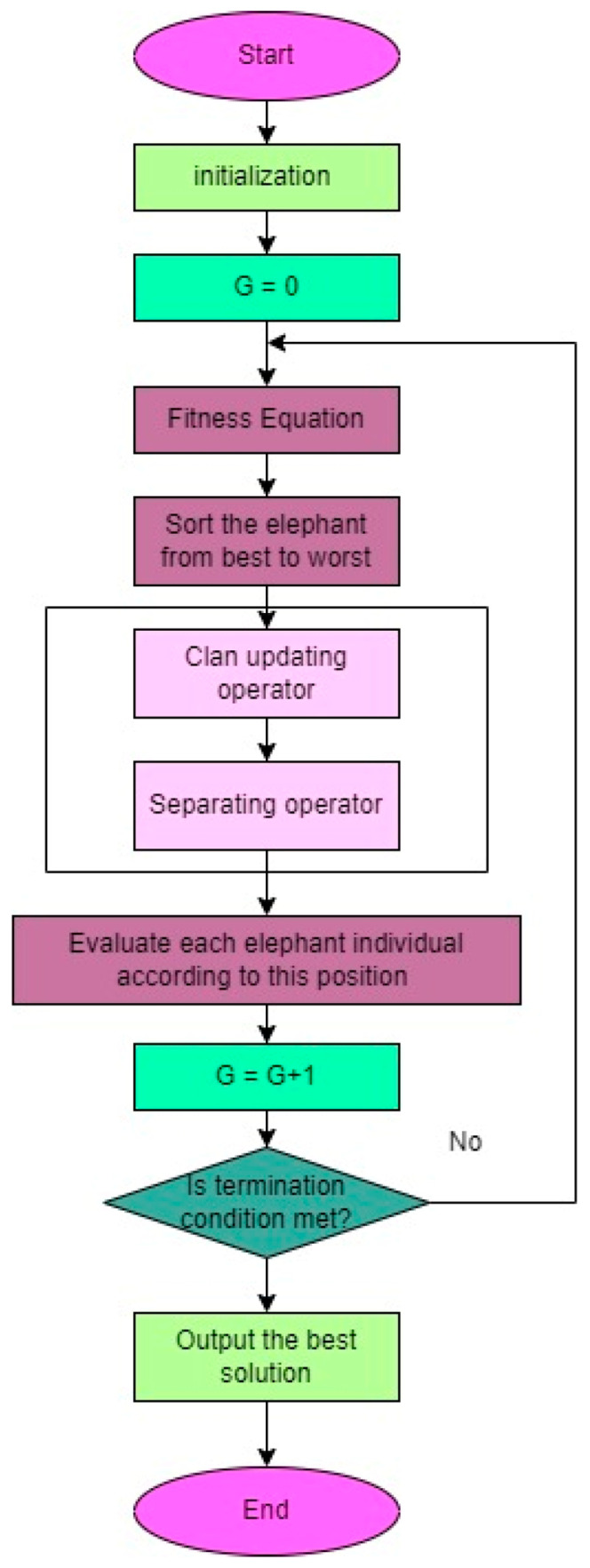
Diagram illustrating the process of the EHO algorithm.

**Figure 11 biomimetics-08-00503-f011:**
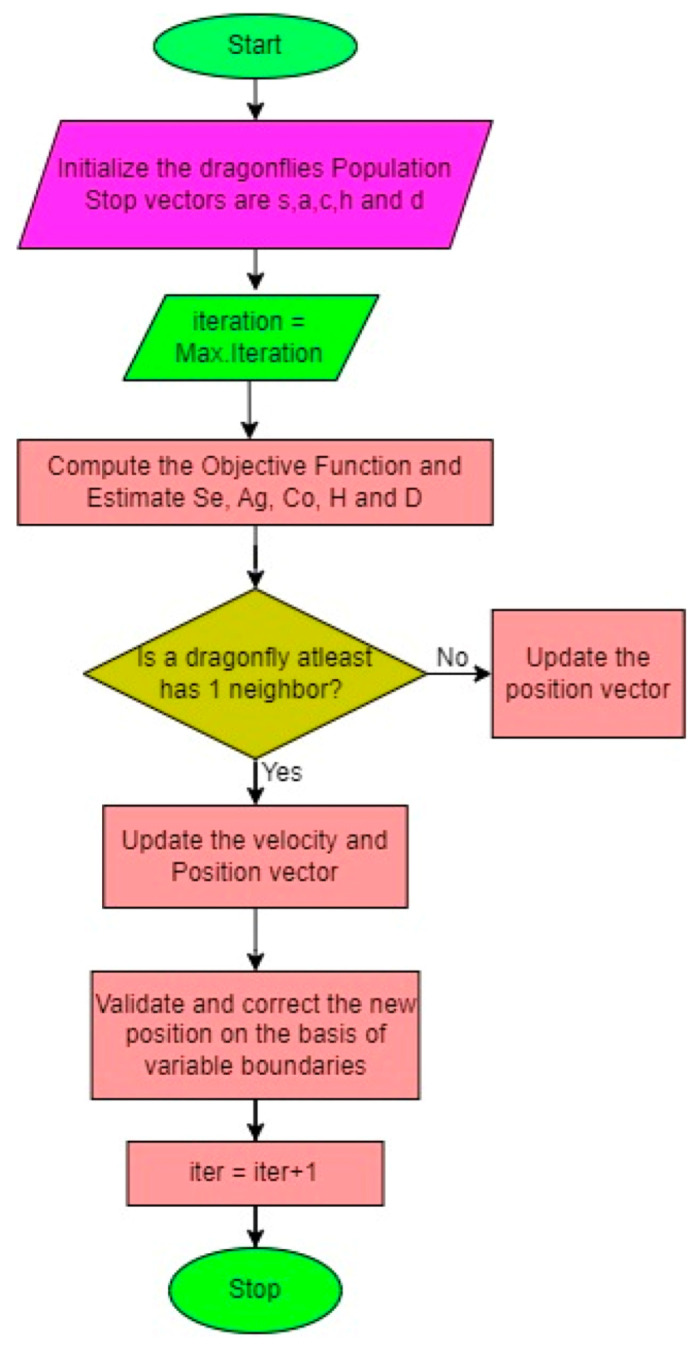
Flowchart of the Dragonfly Optimization algorithm.

**Figure 12 biomimetics-08-00503-f012:**
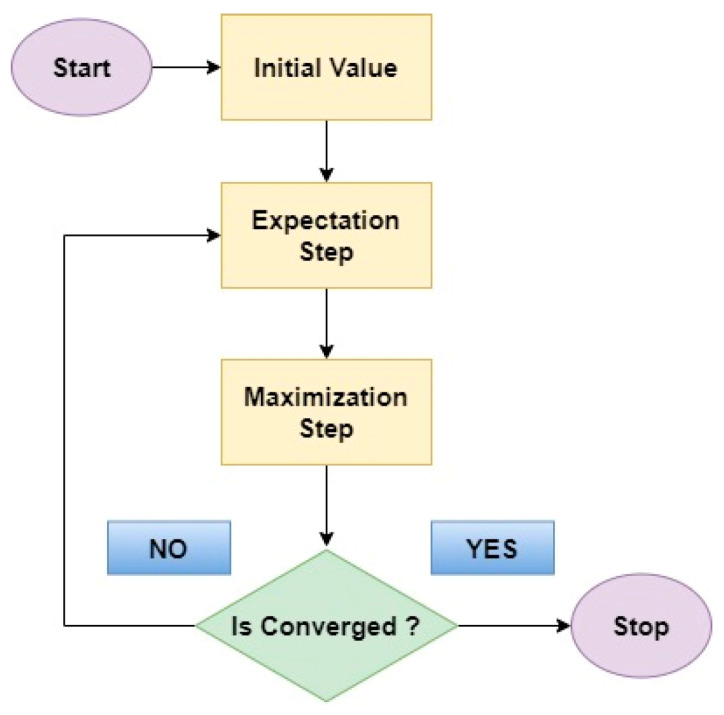
Flow diagram of Expectation Maximum.

**Figure 13 biomimetics-08-00503-f013:**
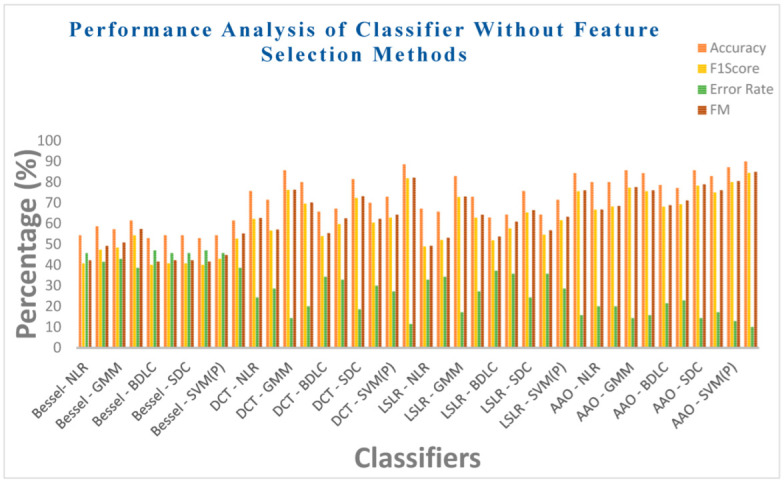
Different classifiers without feature selection methods.

**Figure 14 biomimetics-08-00503-f014:**
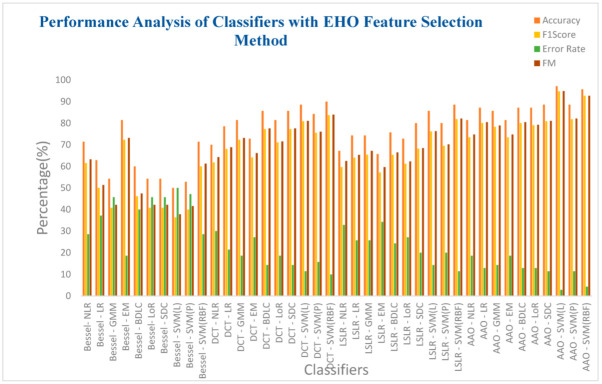
Different classifiers with EHO feature selection methods.

**Figure 15 biomimetics-08-00503-f015:**
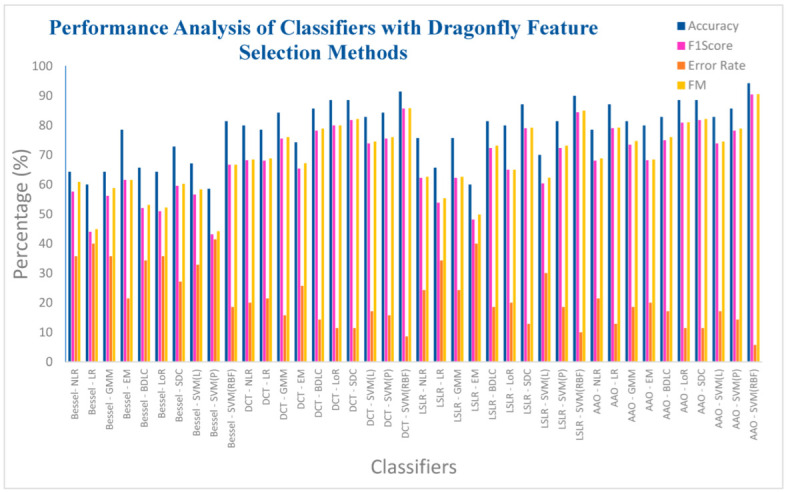
Different classifiers with Dragonfly feature selection method.

**Figure 16 biomimetics-08-00503-f016:**
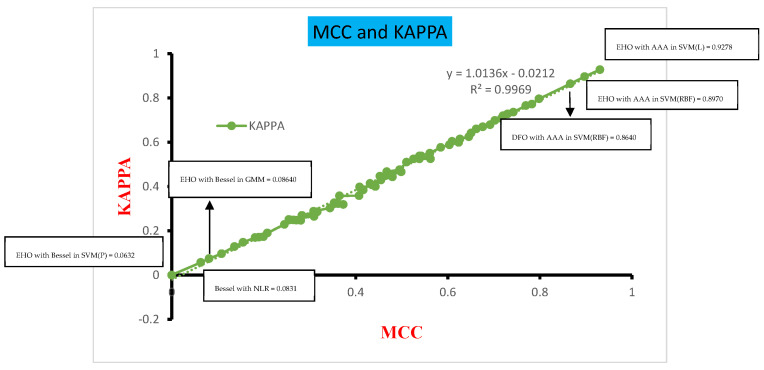
Classifier performance in terms of MCC and Kappa.

**Figure 17 biomimetics-08-00503-f017:**
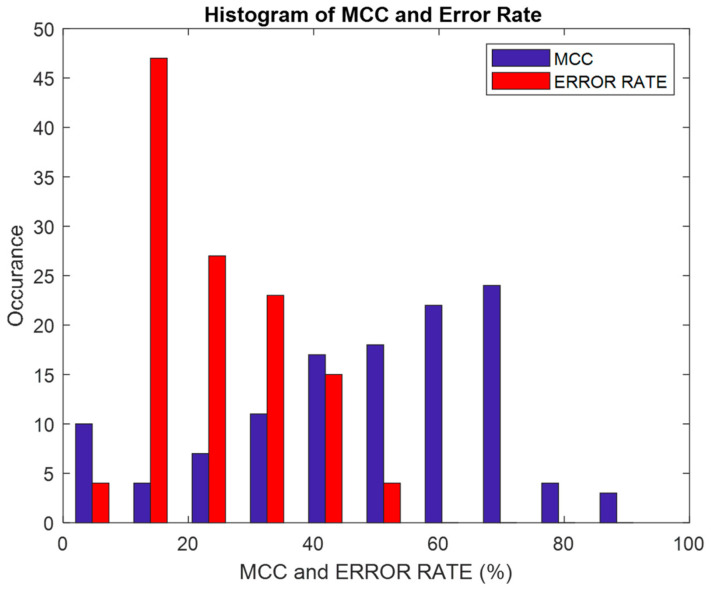
Performance of error rate and MCC (%).

**Table 1 biomimetics-08-00503-t001:** Pancreatic microarray gene dataset for non-diabetic and diabetic classes.

Type	Total Number	Diabetic Class	Non-Diabetic Class	Total Classes
Pancreatic dataset	28,735	20	50	70

**Table 2 biomimetics-08-00503-t002:** Statistical analysis for different DR techniques.

Statistical Parameters	Bessel Function		Discrete Cosine Transform (DCT)		Least Squares Linear Regression (LSLR)		Artificial Algae Algorithm (AAA)	
	Dia	Norm	Dia	Norm	Dia	Norm	Dia	Norm
**Mean**	0.082961	0.084162	1.882012	1.883618	0.00467	0.00457	121.664	120.5492
**Variance**	0.005165	0.005378	0.50819	0.506957	0.000432	0.000417	101.6366	103.0168
**Skewness**	0.865169	0.856162	0.187903	0.228924	0.003787	−0.0315	0.042744	0.054472
**Kurtosis**	0.180926	0.135504	−0.34524	−0.40687	−0.16576	−0.08667	0.152272	0.091169
**Pearson CC**	0.866264	0.859211	0.98138	0.983118	0.975446	0.977318	0.9826	0.985246
**CCA**	0.05904		0.260275		0.090825		0.082321	

**Table 3 biomimetics-08-00503-t003:** Significance of *p*-values for feature selection methods using *t*-test across various DR techniques.

Feature Selection	DR Techniques	Bessel Function		Discrete Cosine Transform (DCT)		Least Squares Linear Regression (LSLR)		Artificial Algae Algorithm (AAA)	
	Genes	Dia	Norm	Dia	Norm	Dia	Norm	Dia	Norm
**EHO**	***p*-value** **< 0.05**	0.9721	0.9998	0.994	0.9996	0.9961	0.9999	0.9466	0.9605
**Dragonfly**	***p*-value** **< 0.05**	0.99985	0.876	0.9956	0.998	0.9951	0.99931	0.9936	0.9977

**Table 5 biomimetics-08-00503-t005:** Analysis of MSE in for different DR techniques without feature selection.

Classifiers	Bessel Function		Discrete Cosine Transform (DCT)		Least Squares Linear Regression (LSLR)		Artificial Algae Algorithm (AAA)	
	MSE Training Set	MSE Testing Set	MSE Training Set	MSE Testing Set	MSE Training Set	MSE Testing Set	MSE Training Set	MSE Testing Set
**NLR**	2.3 × 10^−6^	1.76 × 10^−3^	6.41 × 10^−6^	2.48 × 10^−5^	7.75 × 10^−6^	5.12 × 10^−5^	2.91 × 10^−7^	1.6 × 10^−5^
**LR**	2.41 × 10^−5^	9.51 × 10^−5^	7.52 × 10^−6^	3.11 × 10^−5^	2.18 × 10^−7^	4.66 × 10^−5^	3.67 × 10^−8^	1.45 × 10^−5^
**GMM**	2.1 × 10^−5^	1.75 × 10^−4^	5.72 × 10^−7^	6.8 × 10^−6^	3.09 × 10^−7^	1.11 × 10^−5^	3.76 × 10^−6^	5.33 × 10^−5^
**EM**	**1.62 × 10^−7^**	**9.87 × 10^−6^**	2.71 × 10^−6^	1.3 × 10^−5^	9.87 × 10^−7^	1.99 × 10^−5^	8.97 × 10^−9^	7.3 × 10^−6^
**BLDC**	1.4 × 10^−6^	2.53 × 10^−3^	2.86 × 10^−7^	3.94 × 10^−5^	4.74 × 10^−6^	5.28 × 10^−5^	1.43 × 10^−7^	1.64 × 10^−5^
**LoR**	1.2 × 10^−6^	2.89 × 10^−3^	9.47 × 10^−6^	3.58 × 10^−5^	8.69 × 10^−6^	4.54 × 10^−5^	9.26 × 10^−8^	1.45 × 10^−5^
**SDC**	1.9 × 10^−6^	2.03 × 10^−3^	3.66 × 10^−6^	1.07 × 10^−5^	2.47 × 10^−6^	1.86 × 10^−5^	2.31 × 10^−9^	5 × 10^−6^
**SVM (L)**	3.1 × 10^−6^	2.7 × 10^−3^	8.92 × 10^−6^	2.89 × 10^−5^	1.09 × 10^−5^	4.01 × 10^−5^	4.13 × 10^−9^	8.2 × 10^−6^
**SVM (Poly)**	3.6 × 10^−5^	2.11 × 10^−3^	3.36 × 10^−6^	2.11 × 10^−5^	1.29 × 10^−6^	2.85 × 10^−5^	7.84 × 10^−9^	4.69 × 10^−6^
**SVM (RBF)**	**4.16 × 10^−7^**	**8.3 × 10^−5^**	**1.57 × 10^−8^**	**2.41 × 10^−6^**	**3.22 × 10^−8^**	**5.64 × 10^−6^**	**1.93 × 10^−10^**	**1.77 × 10^−8^**

**Table 6 biomimetics-08-00503-t006:** Analysis of MSE performance for classifiers using EHO feature selection methods across different DR techniques.

Classifiers	Bessel Function		Discrete Cosine Transform (DCT)		Least Squares Linear Regression (LSLR)		Artificial Algae Algorithm (AAA)	
	Training MSE	Testing MSE	Training MSE	Testing MSE	Training MSE	Testing MSE	Training MSE	Testing MSE
**NLR**	4.85 × 10^−6^	2.64 × 10^−5^	4.13 × 10^−5^	2.88 × 10^−5^	1.21 × 10^−6^	3.64 × 10^−5^	7.21 × 10^−7^	9.53 × 10^−6^
**LR**	3.62 × 10^−6^	4.79 × 10^−5^	6.92 × 10^−6^	1.35 × 10^−5^	7.72 × 10^−6^	1.96 × 10^−5^	6.98 × 10^−7^	4.23 × 10^−6^
**GMM**	6.13 × 10^−6^	2.26 × 10^−4^	7.63 × 10^−7^	9.22 × 10^−6^	4.57 × 10^−6^	1.39 × 10^−5^	3.81 × 10^−7^	4.52 × 10^−6^
**EM**	**2.19 × 10^−7^**	**1.2 × 10^−6^**	4.39 × 10^−6^	2.25 × 10^−5^	4.81 × 10^−6^	3.92 × 10^−5^	4.67 × 10^−7^	1 × 10^−5^
**BLDC**	4.47 × 10^−6^	6.56 × 10^−5^	7.94 × 10^−7^	5.8 × 10^−5^	3.72 × 10^−6^	1.56 × 10^−5^	3.52 × 10^−7^	3.97 × 10^−6^
**LoR**	3.24 × 10^−6^	2.26 × 10^−4^	3.32 × 10^−6^	1.09 × 10^−5^	8.37 × 10^−6^	2.26 × 10^−5^	7.61 × 10^−8^	3.82 × 10^−6^
**SDC**	9.62 × 10^−6^	2.31 × 10^−4^	9.13 × 10^−7^	4.62 × 10^−5^	4.87 × 10^−6^	1.52 × 10^−5^	9.93 × 10^−8^	3.84 × 10^−6^
**SVM (L)**	4.12 × 10^−5^	5.29 × 10^−4^	8.47 × 10^−7^	4.16 × 10^−6^	1.93 × 10^−8^	9.61 × 10^−6^	1.67 × 10^−8^	3.81 × 10^−6^
**SVM (Poly)**	6.41 × 10^−5^	2.34 × 10^−4^	2.19 × 10^−7^	6.41 × 10^−6^	5.77 × 10^−8^	1.24 × 10^−5^	1.62 × 10^−8^	2.05 × 10^−6^
**SVM (RBF)**	3.72 × 10^−7^	2.56 × 10^−5^	**6.17 × 10^−8^**	**1.35 × 10^−6^**	**6.79 × 10^−9^**	**2.42 × 10^−6^**	**1.99 × 10^−10^**	**2.5 × 10^−8^**

**Table 7 biomimetics-08-00503-t007:** Analysis of MSE in classifiers for various DR techniques with Dragonfly feature selection methods.

Classifiers	Bessel Function		Discrete Cosine Transform (DCT)		Least Squares Linear Regression (LSLR)		Artificial Algae Algorithm (AAA)	
	Training MSE	Testing MSE	Training MSE	Testing MSE	Training MSE	Testing MSE	Training MSE	Testing MSE
**NLR**	3.62 × 10^−6^	4.54 × 10^−5^	4.16 × 10^−6^	1.36 × 10^−5^	8.21 × 10^−6^	2.72 × 10^−5^	3.86 × 10^−6^	1.28 × 10^−5^
**LR**	4.36 × 10^−6^	7.12 × 10^−5^	2.84 × 10^−6^	1.39 × 10^−5^	9.4 × 10^−6^	3.8 × 10^−5^	2.51 × 10^−8^	4.32 × 10^−6^
**GMM**	7.58 × 10^−7^	4.71 × 10^−5^	5.66 × 10^−8^	7.84 × 10^−6^	3.61 × 10^−6^	2.09 × 10^−5^	4.63 × 10^−8^	1.02 × 10^−5^
**EM**	4.79 × 10^−7^	3.31 × 10^−5^	3.79 × 10^−8^	1.68 × 10^−5^	5.33 × 10^−6^	6.12 × 10^−5^	3.43 × 10^−8^	1.46 × 10^−5^
**BLDC**	6.52 × 10^−7^	4.16 × 10^−5^	2.92 × 10^−8^	4.49 × 10^−5^	7.54 × 10^−8^	9.12 × 10^−6^	7.68 × 10^−8^	8.1 × 10^−6^
**LoR**	6.54 × 10^−7^	5.04 × 10^−5^	7.23 × 10^−8^	6.05 × 10^−6^	1.92 × 10^−7^	2.23 × 10^−6^	4.84 × 10^−9^	3.36 × 10^−6^
**SDC**	3.86 × 10^−7^	2.57 × 10^−5^	8.95 × 10^−7^	3.08 × 10^−6^	7.52 × 10^−8^	6.31 × 10^−6^	1.63 × 10^−8^	2.52 × 10^−6^
**SVM (L)**	5.42 × 10^−7^	3.51 × 10^−5^	8.45 × 10^−7^	1.03 × 10^−5^	1.41 × 10^−7^	2.83 × 10^−5^	1.95 × 10^−7^	1.7 × 10^−6^
**SVM (Poly)**	9.67 × 10^−7^	7.23 × 10^−5^	6.67 × 10^−6^	7.08 × 10^−6^	6.3 × 10^−7^	1.05 × 10^−5^	6.42 × 10^−8^	5.33 × 10^−6^
**SVM (RBF)**	**8.64 × 10^−8^**	**2.72 × 10^−6^**	**1.82 × 10^−8^**	**9.05 × 10^−7^**	**3.4 × 10^−8^**	**1.69 × 10^−6^**	**1.66 × 10^−8^**	**3.25 × 10^−8^**

**Table 8 biomimetics-08-00503-t008:** Selection of optimal parametric values for classifiers.

Classifiers	Description
NLR	Uniform weight w = 0.4, bias b = 0.001, iteratively modified sum of least square error, criterion: MSE
Linear Regression	Uniform weight w = 0.451, bias b = 0.003, criterion: MSE
GMM	Mean covariance of the input samples and tuning parameter using EM steps. Criterion: MSE
EM	0.13 likelihood probability, 0.45 cluster probability, with convergence rate of 0.631. Condition: MSE
BDLC	P(y), prior probability: 0.5, class mean: 0.85, 0.1; criterion: MSE
Logistic regression	Threshold H*θ*(x) < 0.48 with criterion: MSE
SDC	Γ = 0.5 along with mean of each class target values as 0.1 and 0.85
SVM (Linear)	C (Regularization Parameter): 0.85, class weights: 0.4, convergence criterion: MSE
SVM (Polynomial)	C: 0.76, coefficient of the kernel function (gamma): 10, class weights: 0.5, convergence criterion: MSE
SVM (RBF)	C: 1, coefficient of the kernel function (gamma): 100, class weights: 0.86, convergence criterion: MSE

**Table 9 biomimetics-08-00503-t009:** Performance metrics.

Metrics	Formula	Assessment Focus
Accuracy	$Acc=\frac{\left(TN+TP\right)}{\left(TN+FN+TP+FP\right)}$	Fraction of predictions that are correct
F1 Score	$F1=\frac{2\times TP}{\left(2\times TP+FP+FN\right)}$	Harmonic mean of precision and recall
Matthews Correlation Coefficient (MCC)	$MCC=\frac{\left(TP\times TN-FP\times FN\right)}{\sqrt{TP+FP)\times (TP+FN)\times (TN+FP)\times (TN+FN)}}$	Correlation between the observed and predicted classifications
Error Rate	$Error\;rate=\frac{\left(FP+FN\right)}{\left(TP+TN+FP+FN\right)}$	Fraction of predictions that are incorrect
FM Metric	$FM=\sqrt{\left(\frac{TP}{TP+FP}\right)\times \left(\frac{TP}{TP+FN}\right)}$	Generalization of the F-measure that adds a beta parameter
Kappa	$Kappa=\frac{{(P}_{o}-P_{e})}{\left(1-P_{e}\right)}\;P_{o}=\frac{\left(TP+TN\right)}{\left(TP+TN+FP+FN\right)}\;P_{e}=\frac{\left(TP+FP)\times (TP+FN)+(FP+TN)\times (FN+TN\right)}{{\left(TP+TN+FP+FN\right)}^{2}}$	Statistic that measures agreement between observed and predicted classifications, adjusted for chance

Abbreviations: *TP*—true positive: an accurate prediction where the true value was positive. *TN*—true negative: an accurate prediction where the true value was negative. *FP*—false positive: an inaccurate prediction where the actual value was negative. *FN*—false negative: an erroneous prediction where the actual value was positive.

**Table 10 biomimetics-08-00503-t010:** Parametric analysis of different classifiers through various DM techniques.

Dimensionality Reduction	Classifiers	Parameters					
		Accuracy (%)	F1 Score (%)	MCC	Error Rate (%)	FM (%)	Kappa
**Bessel Function**	**NLR**	54.2857	40.7407	0.0813	45.7142	42.1831	0.0743
	**LR**	58.5714	47.2727	0.1897	41.4285	49.1354	0.1714
	**GMM**	57.1428	48.2758	0.1995	42.8571	50.7833	0.1732
	**EM**	61.4285	54.2372	0.3092	38.5714	57.2892	0.2645
	**BLDC**	52.8571	40	0.0632	47.1428	41.5761	0.0571
	**LoR**	54.2857	40.7407	0.0813	45.7142	42.1831	0.0743
	**SDC**	54.2857	40.7407	0.0813	45.7142	42.1831	0.0743
	**SVM (L)**	52.8571	40	0.0632	47.1428	41.5761	0.0571
	**SVM (Poly)**	54.2857	42.8571	0.1084	45.7142	44.7214	0.0967
	**SVM (RBF)**	61.4285	52.6315	0.2805	38.5714	55.1411	0.2470
**Discrete Cosine Transform (DCT)**	**NLR**	75.7142	62.2222	0.4525	24.2857	62.6099	0.4465
	**LR**	71.4285	56.5217	0.3646	28.5714	57.0088	0.3577
	**GMM**	85.7142	76.1904	0.6617	14.2857	76.277	0.6601
	**EM**	80	69.5652	0.5609	20	70.1646	0.5504
	**BLDC**	65.7142	53.8461	0.3083	34.2857	55.3399	0.2881
	**LoR**	67.1428	59.6491	0.4072	32.8571	62.4932	0.3585
	**SDC**	81.4285	72.3404	0.6032	18.5714	73.1564	0.5882
	**SVM (L)**	70	60.3773	0.4162	30	62.2799	0.3849
	**SVM (Poly)**	72.8571	62.7451	0.4547	27.1428	64.2575	0.4291
	**SVM (RBF)**	88.5714	81.8181	0.7423	11.4285	82.1584	0.7358
**Least Squares Linear Regression (LSLR)**	**NLR**	67.1428	48.8888	0.2545	32.8571	49.1935	0.2511
	**LR**	65.7142	52	0.2829	34.2857	53.0723	0.2695
	**GMM**	82.8571	72.7272	0.6091	17.1428	73.0297	0.6037
	**EM**	72.8571	62.7451	0.4547	27.1428	64.2575	0.4291
	**BLDC**	62.8571	51.8518	0.2711	37.1428	53.6875	0.2479
	**LoR**	64.2857	57.6271	0.3728	35.7142	60.8698	0.3190
	**SDC**	75.7142	65.3061	0.4952	24.2857	66.4364	0.4757
	**SVM (L)**	64.2857	54.5454	0.3162	35.7142	56.6947	0.2857
	**SVM (Poly)**	71.4285	61.5384	0.4352	28.5714	63.2456	0.4067
	**SVM (RBF)**	84.2857	75.5555	0.6505	15.7142	76.0263	0.6418
**Artificial Algae Algorithm (AAA)**	**NLR**	80	66.6666	0.5254	20	66.7424	0.5242
	**LR**	80	68.1818	0.5424	20	68.4653	0.5377
	**GMM**	85.7142	77.2727	0.6757	14.2857	77.594	0.6698
	**EM**	84.2857	75.5555	0.6505	15.7142	76.0263	0.6418
	**BLDC**	78.5714	68.0851	0.5382	21.4285	68.853	0.5248
	**LoR**	77.1428	69.2307	0.5622	22.8571	71.1512	0.5254
	**SDC**	85.7142	78.2608	0.6918	14.2857	78.9352	0.6788
	**SVM (L)**	82.8571	75	0.6454	17.1428	76.0639	0.625
	**SVM (Poly)**	87.1428	80	0.7165	12.8571	80.4984	0.7069
	**SVM (RBF)**	90	84.4444	0.7825	10	84.9706	0.7720

**Table 11 biomimetics-08-00503-t011:** Performance metrics with Elephant Herding Optimization (EHO) feature selection method for different DR techniques.

Dimensionality Reduction	Classifiers	Parameters					
		Accuracy (%)	F1 Score (%)	MCC	Error Rate (%)	FM (%)	Kappa
**Bessel Function**	**NLR**	71.4285	61.5384	0.4352	28.5714	63.2456	0.4067
	**LR**	62.8571	50	0.2448	37.1428	51.387	0.2288
	**GMM**	54.2857	40.7407	0.0813	45.7142	42.1831	0.0743
	**EM**	81.4285	72.3404	0.6032	18.5714	73.1564	0.5882
	**BLDC**	60	46.1538	0.1813	40	47.4342	0.1694
	**LoR**	54.2857	40.7407	0.0813	45.7142	42.1831	0.0743
	**SDC**	54.2857	40.7407	0.0813	45.7142	42.1831	0.0743
	**SVM (L)**	50	36.3636	0	50	37.7964	0
	**SVM (Poly)**	52.8571	40	0.0632	47.1428	41.5761	0.0571
	**SVM (RBF)**	71.4285	60	0.4107	28.5714	61.2372	0.3913
**Discrete Cosine Transform (DCT)**	**NLR**	70	61.8181	0.4427	30	64.254	0.4
	**LR**	78.5714	68.0851	0.5382	21.4285	68.853	0.5248
	**GMM**	81.4285	72.3404	0.6032	18.5714	73.1564	0.5882
	**EM**	72.8571	64.1509	0.4796	27.1428	66.1724	0.4435
	**BLDC**	85.7142	77.2727	0.6757	14.2857	77.594	0.6698
	**LoR**	81.4285	71.1111	0.5845	18.5714	71.5542	0.5767
	**SDC**	85.7142	77.2727	0.6757	14.2857	77.594	0.6698
	**SVM (L)**	88.5714	80.9523	0.7298	11.4285	81.0443	0.7281
	**SVM (Poly)**	84.2857	75.5555	0.6505	15.7142	76.0263	0.6418
	**SVM (RBF)**	90	83.7209	0.7694	10	83.9254	0.7655
**Least Squares Linear Regression (LSLR)**	**NLR**	67.1428	59.6491	0.4072	32.8571	62.4932	0.3585
	**LR**	74.2857	64	0.4746	25.7142	65.3197	0.4521
	**GMM**	74.2857	65.3846	0.4987	25.7142	67.1984	0.4661
	**EM**	65.7142	57.1428	0.3615	34.2857	59.6285	0.3225
	**BLDC**	75.7142	65.3061	0.4952	24.2857	66.4364	0.4757
	**LoR**	72.8571	61.2244	0.4310	27.1428	62.2841	0.4140
	**SDC**	80	68.1818	0.5424	20	68.4653	0.5377
	**SVM (L)**	85.7142	76.1904	0.6617	14.2857	76.277	0.6601
	**SVM (Poly)**	80	69.5652	0.5609	20	70.1646	0.5504
	**SVM (RBF)**	88.5714	81.8181	0.7423	11.4285	82.1584	0.7358
**Artificial Algae Algorithm (AAA)**	**NLR**	81.4285	73.4693	0.6236	18.5714	74.7409	0.5991
	**LR**	87.1428	80	0.7165	12.8571	80.4984	0.7069
	**GMM**	85.7142	78.2608	0.6918	14.2857	78.9352	0.6788
	**EM**	81.4285	73.4693	0.6236	18.5714	74.7409	0.5991
	**BLDC**	87.1428	80	0.7165	12.8571	80.4984	0.7069
	**LoR**	87.1428	79.0697	0.7021	12.8571	79.2629	0.6985
	**SDC**	88.5714	80.9523	0.7298	11.4285	81.0443	0.7281
	**SVM (L)**	97.1428	94.7368	0.9302	2.85714	94.8683	0.9278
	**SVM (Poly)**	88.5714	81.8181	0.7423	11.4285	82.1584	0.7358
	**SVM (RBF)**	95.7142	92.6829	0.8970	4.28571	92.7105	0.8965

**Table 12 biomimetics-08-00503-t012:** Performance metric of different classifiers with four DR techniques with the Dragonfly feature selection method.

DR	Classifiers	Parameters					
		Accuracy (%)	F1 Score (%)	MCC	Error Rate (%)	FM (%)	Kappa
**Bessel Function**	**NLR**	64.2857	57.6271	0.3728	35.7142	60.8698	0.3190
	**LR**	60	44	0.1551	40	44.9073	0.1478
	**GMM**	64.2857	56.1403	0.3438	35.7142	58.8172	0.3027
	**EM**	78.5714	61.5384	0.4673	21.4285	61.5587	0.4670
	**BLDC**	65.7142	52	0.2829	34.2857	53.0723	0.2695
	**LoR**	64.2857	50.9803	0.2637	35.7142	52.2093	0.2489
	**SDC**	72.8571	59.5744	0.4083	27.1428	60.2464	0.3981
	**SVM (L)**	67.1428	56.6037	0.3529	32.8571	58.3874	0.3263
	**SVM (Poly)**	58.5714	43.1372	0.1364	41.4285	44.1771	0.1287
	**SVM (RBF)**	81.4285	66.6666	0.5384	18.5714	66.6886	0.5380
**Discrete Cosine Transform (DCT)**	**NLR**	80	68.1818	0.5424	20	68.4653	0.5377
	**LR**	78.5714	68.0851	0.5382	21.4285	68.853	0.5248
	**GMM**	84.2857	75.5555	0.6505	15.7142	76.0263	0.6418
	**EM**	74.2857	65.3846	0.4987	25.7142	67.1984	0.4661
	**BLDC**	85.7142	78.2608	0.6918	14.2857	78.9352	0.6788
	**LoR**	88.5714	80	0.72	11.4285	80	0.72
	**SDC**	88.5714	81.8181	0.7423	11.4285	82.1584	0.7358
	**SVM (L)**	82.8571	73.9130	0.6264	17.1428	74.5499	0.6146
	**SVM (Poly)**	84.2857	75.5555	0.6505	15.7142	76.0263	0.6418
	**SVM (RBF)**	91.4285	85.7142	0.7979	8.57142	85.8116	0.7961
**Least Squares Linear Regression (LSLR)**	**NLR**	75.7142	62.2222	0.4525	24.2857	62.6099	0.4465
	**LR**	65.7142	53.8461	0.3083	34.2857	55.3399	0.2881
	**GMM**	75.7142	62.2222	0.4525	24.2857	62.6099	0.4465
	**EM**	60	48.1481	0.2078	40	49.8527	0.1900
	**BLDC**	81.4285	72.3404	0.6032	18.5714	73.1564	0.5882
	**LoR**	80	65	0.51	20	65	0.51
	**SDC**	87.1428	79.0697	0.7021	12.8571	79.2629	0.6985
	**SVM (L)**	70	60.3773	0.4162	30	62.2799	0.3849
	**SVM (Poly)**	81.4285	72.3404	0.6032	18.5714	73.1564	0.5882
	**SVM (RBF)**	90	84.4444	0.7825	10	84.9706	0.7720
**Artificial Algae Algorithm (AAA)**	**NLR**	78.5714	68.0851	0.5382	21.4285	68.853	0.5248
	**LR**	87.1428	79.0697	0.7021	12.8571	79.2629	0.6985
	**GMM**	81.4285	73.4693	0.6236	18.5714	74.7409	0.5991
	**EM**	80	68.1818	0.5424	20	68.4653	0.5377
	**BLDC**	82.8571	75	0.6454	17.1428	76.0639	0.625
	**LoR**	88.5714	80.9523	0.7298	11.4285	81.0443	0.7281
	**SDC**	88.5714	81.8181	0.7423	11.4285	82.1584	0.7358
	**SVM (L)**	82.8571	73.9130	0.6264	17.1428	74.5499	0.6146
	**SVM (Poly)**	85.7142	78.2608	0.6918	14.2857	78.9352	0.6788
	**SVM (RBF)**	94.2857	90.4761	0.8660	5.71428	90.5789	0.8640

**Table 13 biomimetics-08-00503-t013:** CC of the classifiers without feature selection methods.

Classifiers	DR Method			
	Bessel Function	Discrete Cosine Transform (DST)	Least Squares Linear Regression (LSLR)	Artificial Algae Algorithm (AAA)
**NLR**	O(n^2^logn)	O(n^2^logn)	O(n^3^log2n)	O(n^3^log4n)
**LR**	O(n^2^logn)	O(n^2^logn)	O(n^3^log2n)	O(n^3^log4n)
**GMM**	O(n^2^log2n)	O(n^2^log2n)	O(n^3^log2n)	O(n^3^log4n)
**EM**	O(n^3^logn)	O(n^3^logn)	O(n^3^log2n)	O(n^3^log4n)
**BLDC**	O(n^3^logn)	O(n^3^logn)	O(2n^3^log2n)	O(2n^3^log4n)
**LoR**	O(2n^2^logn)	O(2n^2^logn)	O(2n^4^log2n)	O(2n^4^log4n)
**SDC**	O(n^3^logn)	O(n^3^logn)	O(n^4^log2n)	O(n^4^log4n)
**SVM (L)**	O(2n^3^logn)	O(2n^3^logn)	O(2n^4^log2n)	O(2n^4^log4n)
**SVM (Poly)**	O(2n^3^log2n)	O(2n^3^log2n)	O(2n^4^log4n)	O(2n^4^log8n)
**SVM (RBF)**	O(2n^4^log2n)	O(2n^4^log2n)	O(2n^5^log4n)	O(2n^5^log8n)

**Table 14 biomimetics-08-00503-t014:** CC of the classifiers with EHO feature selection method.

Classifiers	DR Method			
	Bessel Function	Discrete Cosine Transform (DCT)	Least Squares Linear Regression (LSLR)	Artificial Algae Algorithm (AAA)
**NLR**	O(n^4^logn)	O(n^4^logn)	O(n^5^log2n)	O(n^5^log4n)
**LR**	O(n^4^logn)	O(n^4^logn)	O(n^5^log2n)	O(n^5^log4n)
**GMM**	O(n^4^log2n)	O(n^4^log2n)	O(n^5^log2n)	O(n^5^log4n)
**EM**	O(n^5^logn)	O(n^5^logn)	O(n^5^log2n)	O(n^5^log4n)
**BLDC**	O(n^5^logn)	O(n^5^logn)	O(2n^5^log2n)	O(2n^5^log4n)
**LoR**	O(2n^4^logn)	O(2n^4^logn)	O(2n^5^log2n)	O(2n^5^log4n)
**SDC**	O(n^5^logn)	O(n^5^logn)	O(n^6^log2n)	O(n^6^log4n)
**SVM (L)**	O(2n^5^logn)	O(2n^5^logn)	O(2n^6^log2n)	O(2n^6^log4n)
**SVM (Poly)**	O(2n^5^log2n)	O(2n^5^log2n)	O(2n^6^log4n)	O(2n^6^log8n)
**SVM (RBF)**	O(2n^6^log2n)	O(2n^6^log2n)	O(2n^7^log4n)	O(2n^7^log8n)

**Table 15 biomimetics-08-00503-t015:** CC of the classifiers with Dragonfly feature selection method.

Classifiers	DR Method			
	Bessel Function	Discrete Cosine Transform (DST)	Least Squares Linear Regression (LSLR)	Artificial Algae Algorithm (AAA)
**NLR**	O(4n^3^logn)	O(4n^3^logn)	O(4n^4^log2n)	O(4n^4^log4n)
**LR**	O(4n^3^logn)	O(4n^3^logn)	O(4n^4^log2n)	O(4n^4^log4n)
**GMM**	O(4n^3^log2n)	O(4n^3^log2n)	O(4n4log2n)	O(4n^4^log4n)
**EM**	O(4n^4^logn)	O(4n^4^logn)	O(4n^4^log2n)	O(4n^4^log4n)
**BLDC**	O(4n^4^logn)	O(4n^4^logn)	O(8n^4^log2n)	O(8n^4^log4n)
**LoR**	O(8n^3^logn)	O(8n^3^logn)	O(8n^5^log2n)	O(8n^5^log4n)
**SDC**	O(4n^4^logn)	O(4n^4^logn)	O(4n^5^log2n)	O(4n^5^log4n)
**SVM (L)**	O(8n^4^logn)	O(8n^4^logn)	O(8n^5^log2n)	O(8n^5^log4n)
**SVM (Poly)**	O(8n^4^log2n)	O(8n^4^log2n)	O(8n^5^log4n)	O(8n^5^log8n)
**SVM (RBF)**	O(8n^5^log2n)	O(8n^5^log2n)	O(8n^6^log4n)	O(8n^6^log8n)

**Table 16 biomimetics-08-00503-t016:** Comparison with previous work.

S.No	Author (with Year)	Description of the Population	Data Sampling	Machine Learning Parameter	Accuracy (%)
1.	Maniruzzaman et al., (2017) [53]	PIDD (Pima Indian diabetic dataset)	Cross-validation K2, K4, K5, K10, and JK	LDA, QDA, NB, GPC, SVM, ANN, AB, LoR, DT, RF	ACC: 92
2.	Pham et al., (2017) [54]	Diabetes: 12,000, aged between 18 and 100 Age (mean): 73	Training set—66%; tuning set 17%; test set—17%	RNN, CLST Memory (C-LSTM)	ACC—79
3.	Hertroijs et al., (2018) [55]	Total: 105,814 Age (mean): greater than 18	Training set of 90% and test set of 10% fivefold cross-validation	Latent Growth Mixture Modeling (LGMM)	ACC: 92.3
4.	ArellanoCampos et al., (2019) [56]	Base L: 7636 follow: 6144 diabetes: 331 age: 32–54	K = 10, cross-validation and bootstrapping model	Cox proportional hazard regression	ACC: 75
5.	Deo et al., (2019) [57]	Total: 140 diabetes: 14 imbalanced age: 12–90	Training set of 70% and 30% test set with fivefold cross-validation, holdout validation	BT, SVM (L)	ACC: 91
6.	Choi et al., (2019) [58]	Total: 8454 diabetes: 404 age: 40–72	Tenfold cross-validation	LoR, LDA, QDA, KNN	ACC: 78, 77 76, 77
7.	Akula et al., (2019) [59]	PIDD Practice Fusion Dataset total: 10,000 age: 18–80	Training set: 800; test set: 10,000	KNN, SVM, DT, RF, GB, NN, NB	ACC: 86
8.	Xie et al., (2019) [60]	Total: 138,146 diabetes: 20,467 age: 30–80	Training set is around 67%, test set is around 33%	SVM, DT, LoR, RF, NN, NB	ACC: 81, 74, 81, 79, 82, 78
9.	Bernardini et al., (2020) [61]	Total: 252 diabetes: 252 age: 54–72	Tenfold cross-validation	Multiple instance learning boosting	ACC: 83
10.	Zhang et al., (2020) [62]	Total: 36,652 age: 18–79	Tenfold cross-validation	LoR, classification, and regression tree, GB, ANN, RF, SVM	ACC: 75, 80, 81, 74, 86, 76
11.	Jain et al., (2020) [63]	Control: 500 diabetes: 268 age: 21–81	Training set is around 70%, test set is around 30%	SVM, RF, k-NN	ACC: 74, 74, 76
12.	Kalagotla et al., (2021) [64]	Pima Indian dataset	Hold out k-fold cross-validation	Stacking multi-layer perceptron, SVM, LoR	ACC: 78
13.	Haneef et al., (2021) [65]	Total 44,659 age 18–69 data are imbalanced	Training set 80%, test set 20%	LDA	ACC: 67
14.	Deberneh et al., (2021) [66]	Total: 535,169, diabetes: 4.3% prediabetes: 36%, age: 18–108	Tenfold cross-validation	RF, SVM, XGBoost	ACC: 73, 73, 72
15.	Zhang et al., (2021) [67]	Total: 37,730, diabetes: 9.4% age: 50–70 imbalanced	Training set is around 80% test set is around 20% Tenfold cross-validation	Bagging boosting, GBT, RF, GBM	ACC: 82
16.	This article	Nordic Islet Transplantation program	Tenfold cross-validation	Bessel function, DCT, LSLR and AAA	95

LDA—Linear Discriminant Analysis; QDA—Quadratic Discriminant Analysis; NB—Naïve Bayes; GPC—Gaussian Process Classification; SVM—Support Vector Machine; ANN—Artificial Neural Network; AB—ADA Boost; LoR—Logistic Regression; DT—Decision Tree; RF—Random Forest; RRN—Recurrent Neural Network; CLST Memory—Convolutional Long Short-Term Memory; BT—Bagged Tree; KNN—k-Nearest Neighbor; GB—Gradient Boost; NN—Neural Network; k-NN—k-Nearest Neighbor; GBT—Bagging Boost GBT; ACC—accuracy.

## Data Availability

The data that support the findings of this study are available from the corresponding author upon reasonable request.

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
