# Peer review of "Enhancement of Classifier Performance Using Swarm Intelligence in Detection of Diabetes from Pancreatic Microarray Gene Data"

_biomimetics, 2023, doi:10.3390/biomimetics8060503_

Round 1

Reviewer 1 Report

This study discusses the use of gene microarray data existing method to detect diabetes, involves the high-dimensional data processing, dimension reduction technique, the integrated application of feature selection and classification technology. This is a potential research direction, but before provide detailed revision suggestion, I would like to highlight some key problems and improve the points:

1. The abstract needs to more clearly summarize the purpose, methodology, main results and conclusions of the study. The current abstract is a bit lengthy and needs to be more concise.

2. Emphasize the importance and potential application of the study to make it easier for the reader to understand its value.

3. In the introduction, the author should describe more relevant research achievements in recent years. I suggest the authors to introduce some recently proposed meta-heuristics such as the monarch butterfly optimization (MBO), slime mould algorithm (SMA), moth search algorithm (MSA), hunger games search (HGS), Runge Kutta method (RUN), colony predation algorithm (CPA), weighted mean of vectors (INFO) and Harris hawks optimization (HHO), rime optimization algorithm(RIME)to make the paper get more readers. And the authors should cite the original references to guide the readers to the right direction.

4. I suggest the authors make a comprehensive investigation of the optimization methods in the literature in the introduction part and give the analysis to the existing works such as (https://doi.org/10.1016/j.compbiomed.2021.104968,https://doi.org/10.1016/j.compbiomed.2021.104984,https://doi.org/10.1016/j.compbiomed.2021.104712,https://doi.org/10.1007/s42235-022-00297-8, https://doi.org/10.1016/j.compbiomed.2021.104558) to make the whole work more in-depth.

5. The introduction needs to set out more clearly the context and importance of the study. Why is detecting diabetes a critical issue? Why use pancreatic microarray genetic data for diabetes detection? Providing more contextual information could help the reader better understand the motivation for the study.

6. The introduction needs to set out more clearly the context and importance of the study. Why is detecting diabetes a critical issue? Why use pancreatic microarray genetic data for diabetes detection? Providing more contextual information could help the reader better understand the motivation for the study.

7. I have questioned the structure of the manuscript. The revised manuscript has an introduction as follows: (1. Introduction, 1.1. Genesis of Diagnosis of Diabetes Through Microarray Gene Technology , 1.2. Review of literature. In my view this structure remains a concern and the manuscript introduction (1 - 1.3) required rewriting with:

a)An Introduction (1.0) setting out the background, motivation, problem statement, claimed contribution, a brief overview of the proposed approach, and a paper structure;

b)A Literature review (2.0) setting out the review of related studies with a comprehensive comparative analysis;

c)Other detailed discussion(s) included in the current introduction (Section 1) need relocating is dedicated sections where M&M and related topics are considered and discussed.

8. Provide information on the rationale for model selection, why the combination of AAA and SVM (RBF) was chosen, and the reasons for the selection of EHO and DOA.

9. In the experimental section, you mentioned the accuracy results for different classifiers and feature selection methods. However, you need to provide more details such as experimental setup, parameter configurations and cross-validation. This will help the reader to understand the reproducibility and robustness of the experiments.

10. The accuracy of AAA vs SVM (RBF) is mentioned in the results, but no in-depth analysis or explanation is provided. Please discuss why AAA with SVM (RBF) performs well without feature selection and why performance improves after feature selection is applied.

11. Interpretation and discussion of experimental results needs to be more in-depth. You need to explain why certain methods perform better after feature selection and what these results mean in clinical practice. Also, consider performing statistical analyses to support your findings.

12. Discuss the practical applications and potential limitations of the experimental results. In what situations can your method be used? What are the limitations or room for improvement?

13. I noted the brief paragraph addressing future work in the Conclusion Section. However, this failed to consider open research questions (ORQ) identified in the study along with related directions for future research. This is important and must be provided.

14. The conclusion should summarize the key findings and contributions of the paper. It would be helpful to restate the achieved prediction accuracy, discuss the implications of the results, and highlight potential future research directions.

This study discusses the use of gene microarray data existing method to detect diabetes, involves the high-dimensional data processing, dimension reduction technique, the integrated application of feature selection and classification technology. This is a potential research direction, but before provide detailed revision suggestion, I would like to highlight some key problems and improve the points:

1. The abstract needs to more clearly summarize the purpose, methodology, main results and conclusions of the study. The current abstract is a bit lengthy and needs to be more concise.

2. Emphasize the importance and potential application of the study to make it easier for the reader to understand its value.

3. In the introduction, the author should describe more relevant research achievements in recent years. I suggest the authors to introduce some recently proposed meta-heuristics such as the monarch butterfly optimization (MBO), slime mould algorithm (SMA), moth search algorithm (MSA), hunger games search (HGS), Runge Kutta method (RUN), colony predation algorithm (CPA), weighted mean of vectors (INFO) and Harris hawks optimization (HHO), rime optimization algorithm(RIME)to make the paper get more readers. And the authors should cite the original references to guide the readers to the right direction.

4. I suggest the authors make a comprehensive investigation of the optimization methods in the literature in the introduction part and give the analysis to the existing works such as (https://doi.org/10.1016/j.compbiomed.2021.104968,https://doi.org/10.1016/j.compbiomed.2021.104984,https://doi.org/10.1016/j.compbiomed.2021.104712,https://doi.org/10.1007/s42235-022-00297-8, https://doi.org/10.1016/j.compbiomed.2021.104558) to make the whole work more in-depth.

5. The introduction needs to set out more clearly the context and importance of the study. Why is detecting diabetes a critical issue? Why use pancreatic microarray genetic data for diabetes detection? Providing more contextual information could help the reader better understand the motivation for the study.

6. The introduction needs to set out more clearly the context and importance of the study. Why is detecting diabetes a critical issue? Why use pancreatic microarray genetic data for diabetes detection? Providing more contextual information could help the reader better understand the motivation for the study.

7. I have questioned the structure of the manuscript. The revised manuscript has an introduction as follows: (1. Introduction, 1.1. Genesis of Diagnosis of Diabetes Through Microarray Gene Technology , 1.2. Review of literature. In my view this structure remains a concern and the manuscript introduction (1 - 1.3) required rewriting with:

a)An Introduction (1.0) setting out the background, motivation, problem statement, claimed contribution, a brief overview of the proposed approach, and a paper structure;

b)A Literature review (2.0) setting out the review of related studies with a comprehensive comparative analysis;

c)Other detailed discussion(s) included in the current introduction (Section 1) need relocating is dedicated sections where M&M and related topics are considered and discussed.

8. Provide information on the rationale for model selection, why the combination of AAA and SVM (RBF) was chosen, and the reasons for the selection of EHO and DOA.

9. In the experimental section, you mentioned the accuracy results for different classifiers and feature selection methods. However, you need to provide more details such as experimental setup, parameter configurations and cross-validation. This will help the reader to understand the reproducibility and robustness of the experiments.

10. The accuracy of AAA vs SVM (RBF) is mentioned in the results, but no in-depth analysis or explanation is provided. Please discuss why AAA with SVM (RBF) performs well without feature selection and why performance improves after feature selection is applied.

11. Interpretation and discussion of experimental results needs to be more in-depth. You need to explain why certain methods perform better after feature selection and what these results mean in clinical practice. Also, consider performing statistical analyses to support your findings.

12. Discuss the practical applications and potential limitations of the experimental results. In what situations can your method be used? What are the limitations or room for improvement?

13. I noted the brief paragraph addressing future work in the Conclusion Section. However, this failed to consider open research questions (ORQ) identified in the study along with related directions for future research. This is important and must be provided.

14. The conclusion should summarize the key findings and contributions of the paper. It would be helpful to restate the achieved prediction accuracy, discuss the implications of the results, and highlight potential future research directions.

Author Response

Dear Reviewer

Reviewer 2 Report

This is a very long article. My opinion is that describing all the methods tested to get these results is not really necessary. Maybe it would be appropriate for a tutorial or a speech in front of students. A short description and reference articles would be enough.

I did not understand from the article how microarray gene data are used. How is information about a gene presented? Is it a number? What is the range of the descriptor for each gene? Also, while each method used is described in detail, there is almost no comments on how gene information is presented at the input of each method.

There is no indication of the hardware and software used to compute such an extensive number of methods for dimensionality reduction and for classification. Only one person is responsible for computations, which is a huge task in the case of this article.

There are a lot of errors in the text, some paragraphs are not well placed and others repeat almost the same information. I marked such things with yellow in the PDF file attached to this review. There are so many that it is almost impossible for me to discuss each occurrence.

The quality is not so bad, but there is negligence in editing this article and it should be reviewed in a very detailed way.

Author Response

Dear Reviewer

Round 2

Reviewer 2 Report

Very good job of the English rewriting of this paper!